**Data Availability Statement:** Data available at Figshare database https://doi.org/10.5522/04/21679619.v1.

# The Theory of Planned Behaviour doesn't reveal 'attitude-behaviour' gap? Contrasting the effects of moral norms vs. idealism and relativism in predicting pro-environmental behaviours

**Laura Zaikauskaitė** [ID]*◉, **Alicja Grzybek**◉, **Rachel E. Mumford**◉, **Dimitrios Tsivrikos**◉

Department of Clinical, Educational and Health Psychology, University College London, London, United Kingdom

◉ These authors contributed equally to this work.

* l.zaikauskaite@ucl.ac.uk

## Abstract

The inconsistency between pro-environmental attitudes and behaviours called the 'attitude-behaviour' gap, has been reported in many scenarios relating to sustainable actions. However, the reasons for it are not entirely clear. It has been proposed that the 'attitude-behaviour' gap is driven by the moral dimension whereby moral attitudes fail to translate into subsequent moral behaviours. If so, the lacking integration of moral dimension into the environmental domain serves as a generalisable factor to improve the accuracy of predicting pro-environmental behaviours. Hence, we aimed to explore (i) whether the addition of a moral element to the Theory of Planned Behaviour (TPB) shows the 'attitude-behaviour' gap and (ii) whether the ethics position questionnaire (EPQ) is a more suitable measure of morality within the TPB framework, as compared to that of moral norms. The results from 181 US MTurk participants disclosed that the addition of the moral element to the TPB framework did not reveal the presence of the 'attitude-behaviour' gap, despite both moral norms and idealism significantly predicting pro-environmental attitudes. The findings do not indicate whether moral norms or idealism should be used as a more accurate measure of morality within the TPB framework, although relativism was found to have no significant effects. Further investigation of why the moral element does not reveal the 'attitude-behaviour' gap within the TPB framework predicting pro-environmental behaviours would help understand the reasons why rational choice models tend to overestimate theoretical vs. real-life engagement with sustainability.

## Introduction

The current challenge in predicting pro-environmental behaviours relate to the so-called 'attitude-behaviour' gap–the discrepancy between what people say and do. The 'attitude-

**Funding:** This work was funded by the Department of Experimental Psychology as part of the Undergraduate Student Labs of University College London. Initials of the authors who received each award: LZ The funders had no role in study design, data collection and analysis, decision to publish, or preparation of the manuscript.

**Competing interests:** The authors have declared that no competing interests exist.

behaviour' gap contradicts the well-established theoretical conceptualisation that positive attitudes lead to positive behaviours and vice versa. This discrepant pattern is also evident in other cases of ethical consumption but is especially pronounced in scenarios related to pro-environmental decision-making. Despite the numerous attempts to explain the reasons for the occurrence of the 'attitude-behaviour' gap, the more generalisable reasons are not yet established. As such, few recent studies have proposed morality as the core dimension in predicting the 'attitude-behaviour' gap. Although the authors found supportive evidence, the way morality fits into the rational choice frameworks requires further clarification. Hence, we aim to answer the question (i) whether the 'attitude-behaviour' gap occurs between attitude and intention or intention and behaviour and (ii) whether the moral dimension is already reflected by the variables of the rational choice model. We have chosen to meet our aims by addressing the 'attitude-behaviour' gap issue using the Theory of Planned Behaviour [1]. We believe this framework is the most suitable because it allows adding a moral dimension as an additional predictor of behaviour, thus making it possible to compare the predictive capacity of the model with and without the addition of the moral component. Furthermore, it allows testing the effectiveness of different moral predictors, enabling assessing whether predictive power is impacted by the choice of the moral variable. To the best of our knowledge, none of the studies have various moral measures to assess the predictive capacity of the Theory of Planned Behaviour in the environmental domain.

## Common rational choice models used to assess morality and the 'attitude-behaviour' gap in the environmental domain

The common rational choice models that were used to investigate how morality fits within the rational choice frameworks include the Theory of Planned Behaviour (TPB) [1], Value-Belief-Norm (VBN) [2], and the General Theory of Marketing Ethics (GTME) [3, 4]. Of the three, the most distinct model is that of GTME because it assumes a moral component as central rather than an addition predictor of environmental behaviour [3, 4]. Specifically, GTME is centered upon moral philosophy and the manipulation of its dimensions, deontology, and teleology. The common method to assess the effects of moral dimension includes the incorporation of dilemmas, which manipulate the valence of behaviour (deontology) and its consequences (teleology) [5–11]. According to the model, deontology and teleology predict the ethical evaluation of behaviour, and ethical evaluation leads to the intention to perform the behaviour and the actual behaviour [3, 4]. Out of the few studies that used GTME to predict the behaviours in the environmental domain, note: the following studies did not incorporate the dilemmas [12–14], the most relevant is the recent study performed by Zaikauskaite and colleagues [15], note: incorporates the dilemmas. Here, the authors have found that deontology contributed to predicting ethical evaluation more strongly than teleology, and these findings are in line with the previous GTME studies in other than environmental domains [15]. However, the authors failed to find the discriminant validity between pro-environmental intentions and behaviours, suggesting no distinction between the two constructs. Although the authors concluded that the 'attitude-behaviour' gap seemed to occur between attitudes (ethical evaluation) and intentions rather than intentions and behaviours, it's still unclear whether this finding is not biased by the methodological shortcoming of failing to find the distinction between the two outcome variables. Hence, the question of whether the moral dimension fails to translate from attitudes to intentions or from intentions to behaviours requires further evidence.

The more commonly used model to predict environmental behaviour is that of VBN [16–21]. In fact, VBN is the only model specifically designed to predict non-activist

environmentalism [2]. According to VBN, intentions and behaviours are predicted by a chain of variables relating to values (biospheric, altruistic, egoistic), beliefs (ecological worldview, awareness of consequences, ascription of responsibility), and personal norms. Here, the measurement of morality is integrated within personal norms that assess one's moral obligation to perform environmental actions. Hence, the conceptualisation suggests that morality is already integrated within the previous variables of VBN (i.e., values and beliefs). To this end, such a structure barely helps in isolating the effects of the moral dimension. It makes it difficult to track how the moral component translates through attitudes to intentions and through intentions to behaviours.

Furthermore, VBN was critiqued for failing to demonstrate a sufficient model fit [22], suggesting shortcomings in either the conceptualisation of the model or its measurement methods. In fact, the empirical analysis of the model showcased that the variable of personal norms was the only significant predictor of environmental behaviour [22], hence supporting the theorisation that morality plays a significant role in driving pro-environmental actions. Yet, exact evidence of how morality integrates into pro-environmental decision-making is yet to be obtained.

The model better suited to isolating the effects of the moral dimension is that of TPB [1]. According to this model, intentions, and behaviours are predicted by three variables, namely attitudes (one's positive or negative evaluation of performing certain behaviour), subjective norms (behaviour-related opinions of other significant people), and perceived behavioural control (refers to one's volitional control of that specific behaviour). In addition, the conceptualisation of the TPB model permits the inclusion of the other variables of interest [23] hence making the comparison of the TPB model with and without the moral component easy to assess. TPB in an environmental domain has often been extended using the variable of moral norms [24–26]. However, the mixed findings request a more systematic research on a number of methodological fronts. For this reason, the effects of morality on the 'attitude-behaviour' gap are still unknown.

## The issues with adding moral norms to the TPB model

Many studies attempted to increase the effectiveness of predicting pro-environmental behaviours by incorporating moral norms into the TPB [27]. Most of the studies found moral norms to be a significant predictor of pro-environmental intention or behaviour [28–36]. The predictive capacities of the models ($R^2$) ranged from 40% to 66% for most of the reported findings [29, 30, 33–36]. However, there were cases when moral norms did not make a significant contribution to predicting outcome variable(s) [37, 38]. In a similar vein, some of the studies found moral norms to increase $R^2$ or the model fit of the TPB [33, 39], whereas others found the opposite even in cases when moral norms contributed to predicting outcome variable(s) significantly [39]: the decrease of $R^2$; or Kaiser and Scheuthle [26], Yazdanpanah and Forouzani [40]: the decrease of the model fit. In contrast, some researchers reported that the inclusion of moral norms has merely contributed to increasing the predictive power of the TPB [26, 37]. Hence, the conditions when moral norms contribute to the significant prediction of pro-environmental outcomes and increased predictive capacity of the TPB model are not entirely clear.

One of the most considerable lines of debates within the moral norms and environmental behaviour literature relates to the question of whether moral norms are distinct from attitudes or whether they are already represented by attitudes and possibly other TPB variables. That is, it could be that morality is already reflected by attitudes or the rest of the TPB variables, and the addition of moral norms only inflates the result. For this reason, some of the studies

suggested replacing attitudes with moral norms [24–26, 33, 39], although others have argued the discriminant validity between the two constructs [28, 29, 31, 32, 34–36]. The rationale and placement of the moral norms within the TPB model require further revision.

The main difficulty in assessing the effectiveness of moral norms across the studies lies in how they were conducted. First, researchers have measured moral norms using various scales. Thus, this makes it difficult to identify whether the difference in the results is due to the change in measuring the constructe.g., some scales are more effective than others [25, 26]. Second, many studies added more than a single variable of moral norms to the TPB model. For example, Lizin and colleagues [32] added moral norms, past behaviour, consequences, lack of habit, and perceived policy effectiveness. Tan and colleagues [36] added moral norms, environmental concerns, and environmental knowledge. Yazdanpanah and Forouzani [40] added moral norms and self-identity. Chen and Tung [29] added moral norms, the consequences of recycling, and the perceived lack of facilities. Donald and colleagues [31] added moral norms, environmental concerns, and habits. Kaiser [25] added moral norms and anticipated feelings of regret, etc. Therefore, the fine line of how moral norms contribute to the TPB model individually is rarely clear.

Third, some of the studies have measured intention [29, 30, 32, 35, 36, 39, 40], whereas other ones have measured behaviour [28, 34]. In fact, only several studies have incorporated both intention and behaviour [24–26, 31, 33, 38]. Indeed, many researchers have previously found intention to be an overestimated factor and provide inflated, possibly socially desirable results, which do not accurately translate into performing the actual behaviours [27, 41, 42]. Therefore, it is difficult to assess whether the alternating (in)effectiveness of moral norms within the TPB model is impacted by the choice of the outcome variable.

Fourth, the studies investigated different pro-environmental behaviours, such as general recycling [24, 28, 29, 33] or battery pack recycling [32], waste separation [34] or conservation behaviour [25], purchasing of energy-efficient appliances [36] or organic food [40], transport mode use [31] or the aggregate pro-environmental behaviours [26, 30]. The choice of investigating the morality of a certain pro-environmental behaviour could alter the results because some behaviours might be more impacted by the moral component than others [43, 44]. Hence, it's unclear whether/how the focus impacts the (in)effectiveness of adding moral norms to the TPB on specific pro-environmental behaviour.

## An alternative to moral norms: Ethics Position Questionnaire

A different way to classify people's moral judgment was proposed by Forsyth [45, 46]. Forsyth suggested predicting people's moral judgment with a moral philosophy-based measure called Ethics Position Questionnaire [45]. The ethics Position Questionnaire measures the degree of idealism and relativism that stem from deontological and teleological moral philosophies, respectively [47, 48]. The ethics Position Questionnaire differs from the conceptualisation of moral norms because it assesses how people evaluate the action in terms of harm (idealism) and consequences (relativism) rather than in terms of normative behaviour (moral norms) [49, 50]. Specifically, idealists believe that their decisions should never cause harm to others [51, 52], whereas relativists believe that their decisions should result in the best outcomes for all the parties involved. Thus, idealists rely on moral absolutes, while relativists are likely to disregard them when necessary [49, 51]. Importantly, the EPQ measure has demonstrated its robust effects in regression analyses [53, 54] and its consistency when classifying people into four ethics positions (absolutists: high idealism, low relativism; situationists: high idealism, high relativism; exceptionalists: low idealism, low relativism; and subjectivists: low idealism, high relativism; [55–58]. Hence, adding the measure of EPQ instead of moral norms to the

TPB model could shed light on several current issues, such as the discriminant validity between attitudes and moral components or the precise positioning of the moral component within the TPB model.

The research on how EPQ relates to the TPB or environmental domain is scarce. Perhaps the most relevant study was conducted by Agag and Colmekcioglu [59], who have added the EPQ measure [45, 46] to the TPB model in order to predict intention to book green hotels as well as their actual bookings. However, the measures assessing moral obligation, justice, Islamic religiosity, perceived benefits, and perceived risk were also added. Hence, the single contribution that EPQ makes to the TPB model is unclear. Although the authors of the study found the discriminant validity between EPQ and attitudes, EPQ's significant effect on predicting outcome variables, an improved model fit (as compared to the original TPB model), and an increase in $R^2$ (from 49.6% to 72.1%), the findings do not indicate how well the addition of EPQ alone would directly predict intention, and whether it could be used to either predict or substitute attitudes. Thus, it's yet unknown whether the addition of EPQ could be superior to the addition of moral norms.

A direct investigation on EPQ's value in predicting moral judgment of harmful social behaviours (e.g., alcoholism, smoking, etc.) vs. harmful environmental behaviours (e.g., emissions, single-use plastic, etc.) was performed by Zaikauskaite and colleagues [58] (note: the authors did not use the model of TPB). Here, the authors found that both EPQ dimensions (idealism and relativism) significantly contributed to predicting moral judgement of harmful social behaviours. However, the power of relativism was found to diminish for behaviours related to the environmental domain. Specifically, moral judgement of harmful environmental behaviours was significantly predicted by idealism and its interaction with relativism, but not relativism alone. Furthermore, relativism was found to completely lose its power when predicting pro-environmental behaviours, while the significance of idealism remained the same. Hence, the authors have suggested that morality may not be fully integrated into the environmental domain. For this reason, it might be that the measures of moral norms are less suitable for capturing the partial impact of pro-environmental morality, whereas EPQ could address this more accurately.

## The current study

The findings from the reviewed research suggest that the measure of moral norms may introduce certain biases to obtaining clear insights into how morality drives environmental decisions. Indeed, it might be that substituting moral norms with a moral measure of different conceptualisations and placing it within a TPB model could provide more accurate results of how morality integrates within the model. Hence, our current study is designed to shed light on the following research questions:

**RQ1**: Does the addition of the moral component to the TPB model increase its predictive power?

**RQ2**: Does the TPB variable of attitudes already represent a moral component within it, or do the variables of attitudes and morality complement the prediction of outcome variables separately?

**RQ3**: Does EPQ prove to be a more suitable measure of morality within the TPB framework?

**RQ4**: Are certain pro-environmental behaviours more driven by morality than others, or are they all equal?

**RQ5**: Does morality drive the attitude-behaviour gap, and, if so, does it occur between attitudes and intentions or intentions and behaviours?

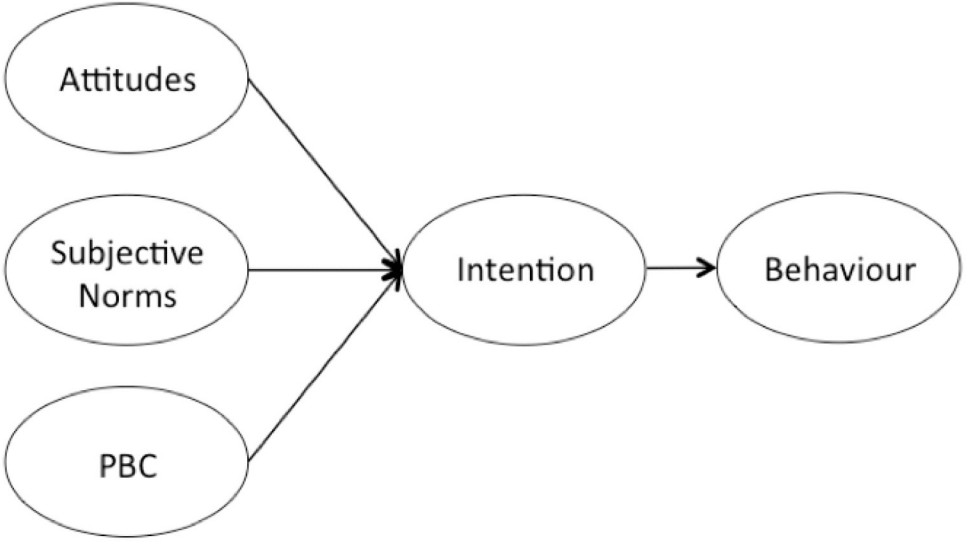

**Fig 1. The original TPB framework.**

We aimed to answer the aforementioned research questions by designing the five variations of the TPB model. First, without adding a moral element to it (Model 1, Fig 1). Second, with adding either moral norms or EPQ as an additional predictor (Models 2 and 3, Fig 2). Third, adding either moral norms or EPQ as a predictor of attitudes (Models 4 and 5, Fig 3). Furthermore, we chose to study 10 pro-environmental behaviours, suggested by Huang [60]. The study set out to provide data for a total of 30 TPB models and an opportunity to aggregate ten behaviours into the three different cases (without the moral element, with moral norms, with

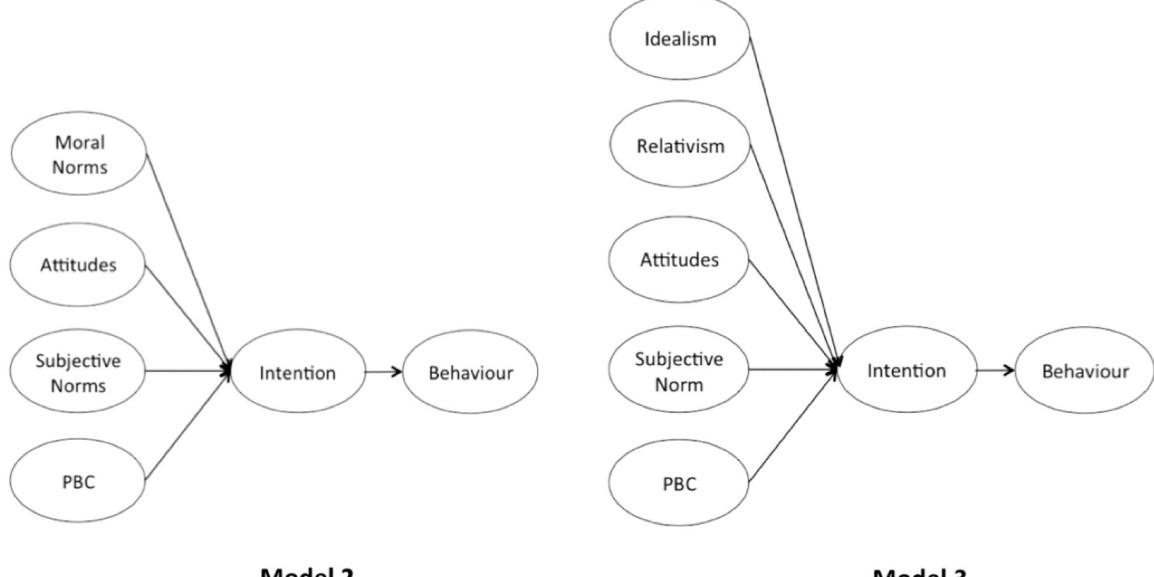

**Fig 2. The TPB frameworks with moral norms and EPQ added as additional predictors.**

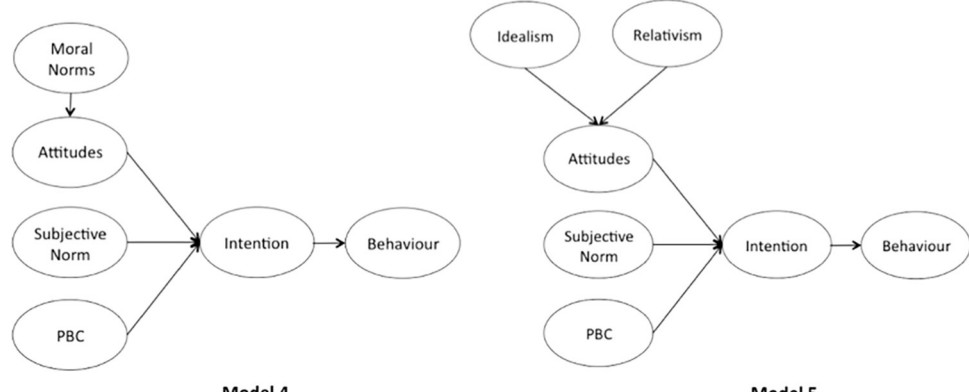

**Fig 3. The TPB frameworks with moral norms and EPQ added as predictors of attitudes.**

EPQ), hence allowing to assess the impact of morality on the individual and the overall set of pro-environmental behaviours. In addition, the inclusion of both intention and behaviour measures is designed to further investigate whether morality drives the 'attitude-behaviour' gap, as suggested by Zaikauskaite and colleagues [15, 58].

## Methods

### Participants

The data from 568 US participants who completed the study from January 2021 –September 2022 was collected using Amazon's Mechanical Turk. The study was divided into six parts and took approx. 1 min (part 1), 1.5 min (parts 2–5), and 2 min (part 6) to complete. Participants were paid $0.60, $1.30, and $1.60, respectively. University College London Ethics Committee granted ethics approval for this study, and all participants gave online consent. All the responses were anonymous, and the researchers had no option to identify individual participants during or after data collection. The results were computed using IBM SPSS v.26 and AMOS v.27.

### Measures

**Pro-environmental behaviours.** Ten pro-environmental behaviour items measuring everyday pro-environmental behaviours, such as recycling, electricity, transportation, etc., were adapted from Huang's study [60]. The frequency of performing presented behaviours was measured on a 7-point Likert scale (e.g., "Recycle newspapers, plastics, and glass," "Compost kitchen waste"; 1-never, 7-every time). Their Cronbach's alpha was 0.81 [60].

**Intention.** Following the technique to transform the behaviour scale into an intention scale [25, 26], we asked participants to rate ten pro-environmental behaviour items [60]. To reduce the error of obtaining socially desirable responses, the intention to perform each of the behaviour was measured using four 7-point items, ranging from strongly disagree to strongly agree (e.g., "I am willing to. . .", "I intend to. . .", "I plan to. . .", "I will. . ."), adapted from the study of Yazdanpanah and Forouzani [40]. Their Cronbach's alpha was 0.88.

**Attitudes.** Similarly, attitudes to perform each of the behaviour were measured using four 7-point items, ranging from strongly disagree to strongly agree (e.g., "I believe that my [recycling behaviour] will help reduce pollution," "I believe that my [recycling behaviour] will help reduce wasteful use of landfills", "I believe that my [recycling behaviour] will help conserve

natural resources", "I feel good about myself when I [recycle]"), adapted from the study of Ramayah and colleagues [61]. Their Cronbach's alpha was 0.78.

**Subjective norm.**   The subjective norm of performing each of the behaviour was measured using four 7-point items, ranging from strongly disagree to strongly agree (e.g., "My friends expect me to [recycle]," "My classmates/colleagues expect me to [recycle]," "Media influences me to [recycle]," "Environmental groups influence me to [recycle]"), adapted from the study of Wan and colleagues [62]. Their Cronbach's alpha was 0.70.

**Perceived behaviour control.**   Perceived behaviour control to perform each of the behaviour was measured using four 7-point items, ranging from strongly disagree to strongly agree (e.g., "I know what [items can be recycled]", "I know where [I can recycle]", "I know how to [recycle]", "I know I would [recycle] if I had more information on [recycling]"), adapted from the study of Wan and colleagues [62]. Their Cronbach's alpha was 0.70.

**Moral norms.**   Moral norms of performing each of the behaviour were measured using four 7-point items, ranging from strongly disagree to strongly agree (e.g., "It would be wrong of me not to [recycle my recyclables]," "I would feel guilty if 'I did not [recycle my recyclables]," "Not [recycling] goes against my principles," "Everybody should share the responsibility to [recycle recyclables]"), adapted from the study of Wan and colleagues [62]. Their Cronbach's alpha was 0.74.

**Moral philosophy.**   Moral philosophy was measured using the Ethics Position Questionnaire (EPQ), which consists of two 10-item scales measuring idealism and relativism [45, 46]. Participants indicated their level of agreement with given statements on a 7-point Likert scale (strongly disagree (1)–strongly agree (7)), with higher scores indicating greater idealism and relativism. The idealism scale included statements such as "People should make certain that their actions never intentionally harm another even to a small degree" and "Risks to another should never be tolerated, irrespective of how small the risks might be." Relativism included statements such as "There are no principles that are so important that they should be a part of any code of ethics" and "What is ethical varies from one situation and society to another." As in previous research [49–51, 53, 54, 56, 63–65], the EPQ dimensions of idealism and relativism revealed a two-factor solution, and Cronbach's alpha was 0.93 for idealism and 0.88 for relativism.

**Attention check items.**   Five attention check items were included in the study. After responding to the second part of the survey (intention), participants were asked to type in three letters which day of the week was February 28. The third part of the survey (attitudes) required participants to type in letters (not numbers) the answer to two times two. The fourth part of the survey (subjective norm) asked the participants to review three answer choices (snow, sun, grass) and indicate which object is typically yellow. The fifth part of the survey presented participants with five numbers and asked them to indicate which one of the answers referred to number 'two.' For the last part of the survey (moral components), participants were presented with a short passage and were asked to move on to the next page once they were done. After this, they were asked to summarise the passage in two to three sentences.

## Procedures

The online survey was launched using the Qualtrics survey platform and set such that all questions on the page needed to be answered before moving on to the next page with questions. The participants completed the study in the web browser. To minimise social desirability bias, which is often present in pro-environmental behaviour studies [66, 67], we have divided the survey into six parts. The first part was conducted four weeks apart from the second one. Each following part became available to the participant once they had completed the previous part.

Part one consisted of a pro-environmental behaviour scale [60], Part two included an intention scale [40], Part three was composed of the scale of attitudes [61], Part four consisted of the scale measuring subjective norms [62], Part 5 included the scale measuring perceived behaviour control [62], and Part 6 included the measures of moral norms [62], moral philosophy [45], and demographics. The data from Part one and Part two of the study were merged using an anonymous response ID, and the merged dataset was utilised for further analyses.

## Results

### Analysis procedures and rationale

First, we ran exploratory factor analysis to check whether questionnaire items loaded on correct factors, indicating their suitability for further multivariate analysis. Second, we ran correlational analyses to assess the strength and directionality of relationships between the study variables. Third, we ran confirmatory factor analysis (CFA) to test the data fit and adjust it for the structural model. As the data were not normally distributed, maximum likelihood with robust standard errors was used for parameter estimation. Based upon Kline's [68] recommendations, the following fit indices were applied: The $X^2/df$ ratio, Root Mean Square Error of Approximation (RMSEA), Standardised Root Mean Residual (SRMR), Comparative Fit Index (CFI), and the Tucker-Lewis Index (TLI). Fourth, we have utilised path analyses to explore the research questions.

### Data cleaning

We have discarded the participants who failed to provide correct answers to the attention check items. Out of 473 complete responses to all six parts of the study, 292 participants failed to provide correct responses to at least one of the six attention check items and thus were removed from the dataset. The final sample consisted of 181 participants.

### Sample demographics

The final sample consisted of 60% males and 39% females (1% preferred not to say). Most of the respondents were 25–34 (43%) and 35–49 (33%) years of age and were either married (with children, 33%) or single (32%). In addition, 53% of the participants were college graduates, 20% reported completing some college, and 19% reported completing post-collegiate. 72% of the respondents were in full-time employment. 23% of the participants reported a level of $50,000-$74,999 annual household income, 20% of the participants reported a level above $75,000, and 19% of the participants reported a level of $40,000-$49,999 annual household income (Table 1).

### Exploratory factor analysis

Exploratory factor analyses (EFA) were run to validate the measures (principal components with a Promax rotation; [69]). We wanted to explore whether each of the ten pro-environmental behaviours was predicted in the same way. Hence, we have divided the questionnaire into ten sets of subsequent items that contributed to predicting each of the ten behaviours. Then, (i) we ran 10 EFAs corresponding to predicting each behaviour. The EFAs included the original TPB variables (Attitudes, Subjective Norm, PBC, Intentions, and Behaviours). Next, (ii) we ran 10 EFAs that included the original TPB variables and moral norms. Last, (iii) we ran 10 EFAs that included the original TPB variables and EPQ. The results of the three sets of EFAs revealed several inconsistencies across the dataset.

**Table 1. Sample demographics ($N$ = 181).**

| Demographics | Item | N | % |
|---|---|---|---|
| Gender | Male | 108 | 60% |
| | Female | 72 | 39% |
| | Prefer not to say | 1 | 1% |
| Age | 18–24 | 6 | 3% |
| | 25–34 | 78 | 43% |
| | 35–49 | 59 | 33% |
| | 50–64 | 30 | 16% |
| | 65 and above | 8 | 4% |
| Marital Status | Single (never married) | 58 | 32% |
| | Married (no children) | 31 | 17% |
| | Married (with children) | 59 | 33% |
| | Domestic partnership | 12 | 6% |
| | Divorced | 1 | 1% |
| | Widowed | 2 | 1% |
| | Separated | 18 | 10% |
| Education | High school or less | 15 | 8% |
| | Some college | 36 | 20% |
| | Undergraduate | 0 | 0% |
| | College graduate | 96 | 53% |
| | Post collegiate | 34 | 19% |
| | None of the above | 0 | 0 |
| Employment Status | Full-time | 130 | 72% |
| | Part-time | 12 | 6% |
| | Self-employed | 14 | 7% |
| | Unemployed | 14 | 7% |
| | Retired | 7 | 4% |
| | Student | 2 | 1% |
| | Other | 2 | 1% |
| Household Income | Less than $9,999 | 4 | 2% |
| | $10,000 - $19,999 | 14 | 8% |
| | $20,000 - $29,999 | 28 | 16% |
| | $30,000 –$39,999 | 22 | 12% |
| | $40,000 - $49,999 | 35 | 19% |
| | $50,000 –$74,999 | 42 | 23% |
| | $75,000 or more | 36 | 20% |

First, intention and behaviour items are loaded onto one factor in most cases. This natural factor solution suggested that behaviour items could not be statistically distinguished from the corresponding intention items, which would suggest that there is no gap between intention and behaviour [15]. Second, there were cases where subjective norms did not load onto correct factors. Third, most of the measures investigating behaviour six (eating less meat), eight (buying energy efficient products), and ten (bringing own utensils when eating out) did not load onto correct factors. Hence, intention and behaviour measures were combined to represent one 'intention-behaviour' factor; subjective norms were excluded from all of the models in order to be able to make meaningful comparisons across the models; behaviours six, eight, and ten were excluded from the further analyses. The final sets of EFAs were forced onto the respective number of factors. Items loading onto incorrect factors, cross-loading, or returning

**Table 2. The results of exploratory factor analysis for the original TPB variables (Model 1).**

| Factors and items | Factor loadings | Communalities |
|---|---|---|
| **Behaviour 1—Recycling** | | |
| *Factor 1: Behaviour-Intention. Cronbach's α = 0.88, Eigenvalue = 6.625, Variance = 44.17%* | | |
| 1.1. Recycle newspapers, plastics, cans, and glass | 0.685 | 0.526 |
| 2.1.1. I am willing to recycle newspapers, plastics, cans, and glass | 0.672 | 0.704 |
| 2.1.2. I intend to recycle newspapers, plastics, cans, and glass | 0.945 | 0.855 |
| 2.1.3. I plan to recycle newspapers, plastics, cans, and glass | 0.891 | 0.760 |
| 2.1.4. I will recycle newspapers, plastics, cans, and glass | 0.849 | 0.797 |
| *Factor 2: Attitudes. Cronbach's α = 0.86, Eigenvalue = 1.602, Variance = 10.68%* | | |
| 3.1.1. I believe that my recycling behaviour will help reduce pollution | 0.869 | 0.768 |
| 3.1.2. I believe that my recycling behaviour will help reduce wasteful use of landfills) | 0.798 | 0.688 |
| 3.1.3. I believe that my recycling behaviour will help conserve natural resources | 0.838 | 0.688 |
| 3.1.4. I feel good about myself when I recycle | 0.793 | 0.728 |
| *Factor 3: Subjective Norm. Cronbach's α = 0.74, Eigenvalue = 1.291, Variance = 8.61%* | | |
| 4.1.1. My friends expect me to recycle recyclables | 0.806 | 0.766 |
| 4.1.2. My classmates/colleagues expect me to recycle recyclables | 0.918 | 0.805 |
| 4.1.3. Media influences me to recycle recyclables | 0.551 | 0.432 |
| *Factor 4: Perceived Behavioural Control. Cronbach's α = 0.81, Eigenvalue = 1.214, Variance = 8.10%* | | |
| 5.1.1. I know what items can be recycled | 0.815 | 0.858 |
| 5.1.2. I know where I can recycle newspapers, plastics, cans, and glass | 0.877 | 0.806 |
| 5.1.3. I know how to recycle my recyclables | 0.803 | 0.704 |
| Total variance = 71.55% | | |
| KMO = 0.875 | | |
| $\chi^2$ = 1546.706 | | |
| df = 105 | | |
| Sig. < 0.001 | | |

factor loadings of less than 0.50 were dropped from the further analysis. The final solutions had internal consistency estimates above 0.60, which is above the limit of acceptability [70]. 21 of the 28 models yielded eigenvalues greater than 1.0. However, seven models resulted in having one of the eigenvalues below the threshold of 1.0 and were within then range of 0.91–0.71, suggesting possible collinearity in the data. Most communalities were above 0.50; however, some items yielded values within the range of 0.39–0.50. The results are presented in Tables 2–5 showing the final factor solutions for behaviour one (recycling) across the four different TPB frameworks, items, consistency estimates, eigenvalues, variance, factor loadings, and community statistics. Factor solutions for the rest of the behaviours could be found in supplementary materials.

## Normality, descriptive statistics, and spearman correlations

Shapiro–Wilk tests were run to test the assumption of normality. Results have revealed that the data was not normally distributed. Further investigation of univariate and multivariate outliers was carried out. Skewness and kurtosis indexes were less than 3 and 10, respectively, and Cook's distances for all the variables were lower than 1 [71]. Thus, the deviation from normality found was not considered severe [72]. Therefore, it was still deemed appropriate to report the means as a measure of central tendency (Table 6).

We have used Spearman's rho to compute the correlation matrix for further CFA and path analysis because this non-parametric measure of association makes no distributional

**Table 3. The results of exploratory factor analysis for the TPB model without subjective norms (adjusted Model 1).**

| Factors and items | Factor loadings | Communalities |
|---|---|---|
| **Behaviour 1—Recycling** | | |
| *Factor 1: Behaviour-Intention. Cronbach's α = 0.88, Eigenvalue = 5.965, Variance = 49.71%* | | |
| 1.1. Recycle newspapers, plastics, cans, and glass | 0.772 | 0.510 |
| 2.1.1. I am willing to recycle newspapers, plastics, cans, and glass | 0.597 | 0.665 |
| 2.1.2. I intend to recycle newspapers, plastics, cans, and glass | 0.923 | 0.850 |
| 2.1.3. I plan to recycle newspapers, plastics, cans, and glass | 0.880 | 0.757 |
| 2.1.4. I will recycle newspapers, plastics, cans, and glass | 0.850 | 0.793 |
| *Factor 2: Attitudes. Cronbach's α = 0.86, Eigenvalue = 1.453, Variance = 12.11%* | | |
| 3.1.1. I believe that my recycling behaviour will help reduce pollution | 0.888 | 0.775 |
| 3.1.2. I believe that my recycling behaviour will help reduce wasteful use of landfills) | 0.801 | 0.675 |
| 3.1.3. I believe that my recycling behaviour will help conserve natural resources | 0.851 | 0.694 |
| 3.1.4. I feel good about myself when I recycle | 0.814 | 0.698 |
| *Factor 3: Perceived Behavioural Control. Cronbach's α = 0.81, Eigenvalue = 1.209, Variance = 10.08%* | | |
| 5.1.1. I know what items can be recycled | 0.821 | 0.682 |
| 5.1.2. I know where I can recycle newspapers, plastics, cans, and glass | 0.895 | 0.778 |
| 5.1.3. I know how to recycle my recyclables | 0.795 | 0.749 |
| Total variance = 71.89% | | |
| KMO = 0.882 | | |
| $\chi^2$ = 1295.529 | | |
| df = 66 | | |
| Sig. < 0.001 | | |

assumptions. This avoids distorting the distribution if there is a reason to believe these characteristics are representative of the underlying population [73]. To test the strength and directionality of the relationships between variables, we ran Spearman's correlations because this test does not assume the parametric distribution of data (Table 7). The results revealed that the correlation between behaviours and relativism was the only non-significant correlation (r = 0.130, P = 0.082). This is in line with the former study by Zaikauskaite and colleagues [58], who also found that relativism was a non-significant predictor of the same pro-environmental behaviours [60].

## Confirmatory factor analysis

CFAs for each of the seven behaviours were subjected to using the three types of models: (i) TPB (without subjective norms), (ii) TPB with moral norms (without subjective norms), and (iii) TPB with EPQ (without subjective norms). Models lacking data fit along the criteria of RMSEA, CFI, and TLI were corrected by deleting the items that were inflating the Chi-square value the most or covarying several error terms [68]. Modified CFA models fitted the data well across the criteria of $X^2/df$, SRMR, and TLI. Three TPB with EPQ models resulted in having CFI values below the recommended criteria of ≥0.95 (recycling: CFI = 0.937; local products: CFI = 0.949; plastic bags: CFI = 0.935), and the rest of the models were above this threshold [74]. Most of the RMSEA values were slightly above the recommended threshold of <0.05 and were within the range of 0.051–0.084 [75]. However, the cut-off of 0.050 has been critiqued by further studies that suggested that the cut-off value of 0.050 is more suitable for the models having larger sample sizes (n>200) and, overall, should not be taken as an absolute 'rule of thumb' [76–80]. Chi-square tests for all except one of the models were significant, which does not align with the expectation. However, it's not uncommon for the models using large data

**Table 4. The results of exploratory factor analysis for TPB extension with moral norms (adjusted Models 2, 4).**

| Factors and items | Factor loadings | Communalities |
|---|---|---|
| **Behaviour 1 –Recycling** | | |
| *Factor 1: Behaviour-Intention. Cronbach's α = 0.88, Eigenvalue = 7.745, Variance = 48.40%* | | |
| 1.1. Recycle newspapers, plastics, cans, and glass | 0.775 | 0.521 |
| 2.1.1. I am willing to recycle newspapers, plastics, cans, and glass | 0.662 | 0.703 |
| 2.1.2. I intend to recycle newspapers, plastics, cans, and glass | 0.917 | 0.851 |
| 2.1.3. I plan to recycle newspapers, plastics, cans, and glass | 0.806 | 0.753 |
| 2.1.4. I will recycle newspapers, plastics, cans, and glass | 0.800 | 0.787 |
| *Factor 2: Attitudes. Cronbach's α = 0.86, Eigenvalue = 1.618, Variance = 10.11%* | | |
| 3.1.1. I believe that my recycling behaviour will help reduce pollution | 0.858 | 0.771 |
| 3.1.2. I believe that my recycling behaviour will help reduce wasteful use of landfills | 0.821 | 0.700 |
| 3.1.3. I believe that my recycling behaviour will help conserve natural resources | 0.861 | 0.707 |
| 3.1.4. I feel good about myself when I recycle | 0.689 | 0.765 |
| *Factor 3: Perceived Behavioural Control. Cronbach's α = 0.81, Eigenvalue = 1.328, Variance = 8.30%* | | |
| 5.1.1. I know what items can be recycled | 0.818 | 0.700 |
| 5.1.2. I know where I can recycle newspapers, plastics, cans, and glass | 0.846 | 0.754 |
| 5.1.3. I know how to recycle my recyclables | 0.765 | 0.733 |
| *Factor 4: Moral Norms. Cronbach's α = 0.89, Eigenvalue = 1.191, Variance = 7.45%* | | |
| 6.1.1. It would be wrong of me **not** to recycle my recyclables | 0.767 | 0.738 |
| 6.1.2. I would feel guilty if I did **not** recycle my recyclables | 0.806 | 0.818 |
| 6.1.3. **Not** recycling goes against my principles | 0.834 | 0.792 |
| 6.1.4. Everybody should share the responsibility to recycle recyclables | 0.911 | 0.789 |
| Total variance = 74.26% | | |
| KMO = 0.902 | | |
| $\chi^2$ = 1927.343 | | |
| df = 120 | | |
| Sig. = 0.000 | | |

samples to achieve significant rather than expected non-significant P-value [68]; therefore, we considered our confirmatory factor model to fit the data adequately well across the overall criteria. Overall, all the other metrics suggested good model fits. Therefore, we have deemed obtained CFI value to be satisfactory for our analysis (Table 8).

## Structural Equation Modeling (SEM)

To test whether the addition of a moral element increases the predictive capacity of the models and reveals the 'attitude-behaviour' gap, the respective items were specified to load onto latent factors. Specifically, (i) we wanted to find out how well the TPB models (without the addition of the moral element) predicted the behaviours. Then, (ii) we wanted to test whether the alteration of the moral element (moral norms vs. EPQ) changed the predictive capacity of the models. Finally, (iii) we wanted to explore whether positioning moral element as an additional predictor vs. predictor of attitudes changed the predictive capacity of the models. Hence, we created five types of models for each of the seven behaviours and subjected them to SEM. Models lacking data fit along the criteria of RMSEA, CFI, and TLI were corrected by deleting the items that were inflating the Chi-square value the most or covarying several error terms [68]. Modified models fitted the data well across the criteria of $X^2/df$, SRMR, and TLI. Eight of the models resulted having CFI values below the recommended criteria of $\geq 0.95$ (values ranged from 0.918–0.947), and the rest of the 20 models showcased CFI values above this

**Table 5. The results of exploratory factor analysis for TPB extension with EPQ (adjusted Models 3, 5).**

| Factors and items | Factor loadings | Communalities |
|---|---|---|
| **Behaviour 1 –Recycling** | | |
| *Factor 1: Behaviour-Intention. Cronbach's α = 0.88, Eigenvalue = 3.410, Variance = 11.00%* | | |
| 1.1. Recycle newspapers, plastics, cans, and glass | 0.783 | 0.529 |
| 2.1.1. I am willing to recycle newspapers, plastics, cans, and glass | 0.591 | 0.677 |
| 2.1.2. I intend to recycle newspapers, plastics, cans, and glass | 0.878 | 0.843 |
| 2.1.3. I plan to recycle newspapers, plastics, cans, and glass | 0.831 | 0.737 |
| 2.1.4. I will recycle newspapers, plastics, cans, and glass | 0.786 | 0.763 |
| *Factor 2: Attitudes. Cronbach's α = 0.86, Eigenvalue = 1.557, Variance = 5.02%* | | |
| 3.1.1. I believe that my recycling behaviour will help reduce pollution | 0.841 | 0.767 |
| 3.1.2. I believe that my recycling behaviour will help reduce wasteful use of landfills | 0.767 | 0.689 |
| 3.1.3. I believe that my recycling behaviour will help conserve natural resources | 0.827 | 0.676 |
| 3.1.4. I feel good about myself when I recycle | 0.742 | 0.684 |
| *Factor 3: Perceived Behavioural Control. Cronbach's α = 0.81, Eigenvalue = 1.315, Variance = 4.24%* | | |
| 5.1.1. I know what items can be recycled | 0.695 | 0.660 |
| 5.1.2. I know where I can recycle newspapers, plastics, cans, and glass | 0.824 | 0.753 |
| 5.1.3. I know how to recycle my recyclables | 0.782 | 0.758 |
| *Factor 4: EPQ: Relativism. Cronbach's α = 0.92, Eigenvalue = 7.961, Variance = 25.68%* | | |
| 7.1.1.There are no ethical principles that are so important that they should be a part of any code of ethics | 0.565 | 0.523 |
| 7.1.2.What is ethical varies from one situation and society to another | 0.689 | 0.502 |
| 7.1.3.Moral standards should be seen as being individualistic; what one person considers to be moral may be judged to be immoral by another person | 0.875 | 0.734 |
| 7.1.4.Different types of moralities cannot be compared as to "rightness" | 0.828 | 0.692 |
| 7.1.5.Questions of what is ethical for everyone can never be resolved since what is moral or immoral is up to the individual | 0.892 | 0.782 |
| 7.1.6.Moral standards are simply personal rules that indicate how a person should behave, which should not be applied in making judgments of others | 0.798 | 0.716 |
| 7.1.7.Ethical considerations in interpersonal relations are so complex that individuals should be allowed to formulate their own individual codes | 0.818 | 0.717 |
| 7.1.8.Rigidly codifying an ethical position that prevents certain types of actions could stand in the way of better human relations and adjustment | 0.747 | 0.560 |
| 7.1.9.No rule concerning lying can be formulated; whether a lie is permissible or not permissible totally depends upon the situation | 0.746 | 0.585 |
| 7.1.10.Whether a lie is judged to be moral or immoral depends upon the circumstances surrounding the action | 0.602 | 0.522 |
| *Factor 5: EPQ: Idealism. Cronbach's α = 0.90, Eigenvalue = 6.064, Variance = 19.56%* | | |
| 7.2.1. A person should make certain that their actions never intentionally harm another, even to a small degree | 0.779 | 0.687 |
| 7.2.2. Risks to another should never be tolerated, irrespective of how small the risks might be | 0.820 | 0.646 |
| 7.2.3. The existence of potential harm to others is always wrong, irrespective of the benefits to be gained | 0.865 | 0.769 |
| 7.2.4. One should never psychologically or physically harm another person | 0.745 | 0.593 |
| 7.2.5. One should not perform an action that might in any way threaten the dignity and welfare of another individual | 0.759 | 0.665 |
| 7.2.6. If an action could harm an innocent other, then it should not be done | 0.747 | 0.630 |
| 7.2.8. The dignity and welfare of the people should be the most important concern in any society | 0.595 | 0.506 |
| 7.2.9. It is never necessary to sacrifice the welfare of others | 0.768 | 0.580 |
| 7.2.10 Moral actions are those which closely match ideals of the most "perfect" action | 0.587 | 0.359 |
| Total variance = 65.51% | | |
| KMO = 0.865 | | |
| $\chi^2$ = 3747.310 | | |
| df = 465 | | |
| Sig. = 0.000 | | |

**Table 6. Descriptives and Shapiro-Wilk test results for normality assumptions ($N$ = 181).**

| Variable | M | SD | Statistic | df | P | Skewness | Kurtosis |
|---|---|---|---|---|---|---|---|
| Behaviour | 4.07 | 1.11 | 0.069 | 181 | 0.155 | 0.125 | -0.396 |
| Intention | 5.15 | 1.03 | 0.620 | 181 | 0.020 | -0.424 | -0.087 |
| Attitudes | 5.29 | 0.93 | 0.101 | 181 | <0.001 | -0.858 | 1.30 |
| Subjective Norms | 4.88 | 1.03 | 0.093 | 181 | 0.002 | -0.462 | 0.021 |
| PBC | 5.27 | 0.71 | 0.096 | 181 | <0.001 | -0.789 | 0.851 |
| Moral Norms | 4.94 | 1.17 | 0.076 | 181 | <0.001 | -0.767 | 0.687 |
| Relativism | 4.15 | 1.06 | 0.076 | 181 | 0.007 | -0.491 | -0.113 |
| Idealism | 5.16 | 0.98 | 0.086 | 181 | <0.001 | -0.688 | 0.660 |

Note: $p$ values above 0.05 are not significant.

threshold [74]. Most of the RMSEA values were slightly above the recommended threshold of <0.050 and were within the range of 0.051–0.081 [75]. Again, the feasibility of having a cut-off value of 0.050 has received a considerable amount of critique [76, 77, 79, 80]. Similarly, Chi-square tests for all the models were significant, which is not in line with the expectation. However, it's not uncommon for the models using large data samples to achieve significant rather than expected non-significant P-value [68]; therefore, we considered our models to fit the data adequately well across the overall criteria. Overall, all the other metrics suggested good model fits. Therefore, we have deemed obtained goodness of fit values for SEM models to be satisfactory for further path analysis (Table 9). Figs 4–6. represent CFA models for behaviour 1 (recycling). CFA models for all 7 behaviours could be found in supplementary materials.

**Table 7. Spearman's correlations and $p$ values for the intercorrelations among study variables ($N$ = 181).**

| | Variable | 1 | 2 | 3 | 4 | 5 | 6 | 7 |
|---|---|---|---|---|---|---|---|---|
| 1 | Behaviours | - | | | | | | |
| | P | - | | | | | | |
| 2 | Intentions | 0.563*** | - | | | | | |
| | P | <0.001 | - | | | | | |
| 3 | Attitudes | 0.387*** | 0.643*** | - | | | | |
| | P | <0.001 | <0.001 | - | | | | |
| 4 | Subjective Norms | 0.417*** | 0.558*** | 0.612*** | - | | | |
| | P | <0.001 | <0.001 | <0.001 | - | | | |
| 5 | PBC | 0.237*** | 0.599*** | 0.604*** | 0.519*** | - | | |
| | P | <0.001 | <0.001 | <0.001 | <0.001 | - | | |
| 6 | Moral Norms | 0.454*** | 0.695*** | 0.703*** | 0.679*** | 0.541*** | - | |
| | P | <0.001 | <0.001 | <0.001 | <0.001 | <0.001 | - | |
| 7 | Relativism | 0.130 | 0.184*** | 0.237** | 0.241** | 0.203** | 0.239** | - |
| | P | 0.082 | 0.013* | 0.001 | 0.001 | 0.006 | 0.001 | - |
| 8 | Idealism | 0.206*** | 0.421*** | 0.477*** | 0.453*** | 0.516*** | 0.598*** | 0.165* |
| | P | <0.001 | <0.001 | <0.001 | <0.001 | <0.001 | <0.001 | 0.026 |

Note:

P* < 0.005

P** < 0.001

P*** < 0.001

**Table 8. Goodness of fit results for the models subjected to CFA (N = 181).**

| Fit index: | $X^2$ | df | $X^2/df$ | P | CFI | RMSEA | SRMR | TLI |
|---|---|---|---|---|---|---|---|---|
| Goodness of fit criterion: | n/a | n/a | <3[a] | >.05[b] | ≥0.95[c] | <0.05[d] | <0.08[c] | >0.80[b] |
| **TPB—SB** | | | | | | | | |
| 1.1. Recycling | 75.20 | 41 | 1.83 | 0.001 | 0.969 | 0.068 [0.043, 0.092] | 0.0376 | 0.958 |
| 1.2. Composting | 60.03 | 31 | 1.94 | 0.001 | 0.981 | 0.072 [0.044, 0.099] | 0.0529 | 0.972 |
| 1.3. Electron. Devices | 70.12 | 40 | 1.75 | 0.002 | 0.976 | 0.065 [0.038, 0.089] | 0.0468 | .0967 |
| 1.4. Air Conditioning | 56.59 | 29 | 1.95 | 0.002 | 0.978 | 0.073 [0.044, 0.101] | 0.0550 | 0.966 |
| 1.5. Transport Use | 54.69 | 39 | 1.40 | 0.049 | 0.987 | 0.047 [0.003, 0.075] | 0.0427 | 0.982 |
| 1.6. Local Products | 70.66 | 31 | 2.28 | 0.084 | 0.964 | 0.084 [0.058, 0.110] | 0.0416 | 0.948 |
| 1.7. Plastic Bags | 71.89 | 41 | 1.75 | 0.002 | 0.965 | 0.065 [0.039, 0.089] | 0.0405 | 0.967 |
| **TPB + Moral Norms** | | | | | | | | |
| 2.1. Recycling | 72.45 | 48 | 1.5 | 0.013 | 0.978 | 0.053 [0.025, 0.077] | 0.0389 | 0.970 |
| 2.2. Composting | 64.50 | 37 | 1.7 | 0.003 | 0.983 | 0.064 [0.037, 0.090] | 0.0376 | 0.975 |
| 2.3. Electron. Devices | 74.70 | 48 | 1.6 | 0.008 | 0.982 | 0.056 [0.029, 0.079] | 0.0342 | 0.975 |
| 2.4. Air Conditioning | 100.46 | 69 | 1.5 | 0.008 | 0.981 | 0.050 [0.026, 0.071] | 0.0532 | 0.975 |
| 2.5. Transport Use | 137.46 | 69 | 2.0 | 0.000 | 0.960 | 0.074 [0.056, 0.092] | 0.0578 | 0.947 |
| 2.6. Local Products | 60.25 | 38 | 1.6 | 0.012 | 0.980 | 0.057 [0.027, 0.083] | 0.0420 | 0.971 |
| 2.7. Plastic Bags | 53.97 | 38 | 1.4 | 0.045 | 0.985 | 0.048 [0.008, 0.076] | 0.0352 | 0.979 |
| **TPB + EPQ** | | | | | | | | |
| 3.1. Recycling | 345.91 | 200 | 1.82 | 0.000 | 0.937 | 0.064 [0.052, 0.075] | 0.0673 | 0.928 |
| 3.2. Composting | 383.96 | 266 | 1.44 | 0.000 | 0.963 | 0.050 [0.030, 0.060] | 0.0574 | 0.958 |
| 3.3. Electron. Devices | 325.88 | 219 | 1.49 | 0.000 | 0.957 | 0.052 [0.040, 0.064] | 0.0527 | 0.950 |
| 3.4. Air Conditioning | 300.77 | 200 | 1.50 | 0.000 | 0.955 | 0.053 [0.040, 0.065] | 0.0544 | 0.948 |
| 3.5. Transport Use | 322.40 | 220 | 1.47 | 0.000 | 0.959 | 0.051 [0.038, 0.062] | 0.0538 | 0.952 |
| 3.6. Local Products | 348.39 | 219 | 1.59 | 0.000 | 0.949 | 0.057 [0.046, 0.068] | 0.0621 | 0.937 |
| 3.7. Plastic Bags | 432.07 | 264 | 1.64 | 0.000 | 0.935 | 0.059 [0.049, 0.069] | 0.0613 | 0.926 |

CFI, Comparative Fit Index; RMSEA, Root Mean Square Error of Approximation [95% CI]; SRMR, Standardised Root Mean Square Residual; TLI, Tucker–Lewis Index.

aValues recommended by van Dam [81].

bValues recommended by Hu and Bentler [78].

cValues recommended by Hooper et al. [74].

dValues recommended by MacCallum et al. [75].

## Path analyses

Path analyses were performed to answer the research questions. First, we wanted to figure out whether the addition of a moral component to the TPB models composed of attitudes, perceived behavioural control, and intention-behaviour measurement items increase their predictive power (RQ1). We have found that the addition of EPQ either as an additional predictor (Model 3) or as a predictor of attitudes (Model 5) has reduced the $R^2$ of the TPB frameworks predicting transport and plastic bag use (as compared to Model 1). The addition of moral norms as a predictor of attitudes to the TPB framework predicting the purchasing of local products (Model 4) has also decreased the primary $R^2$ (as compared to Model 1). However, adding a moral element to the rest of the 23 models revealed an increase in $R^2$, suggesting that the moral element adds value to predicting pro-environmental behaviours (Fig 7). SEM results for Models 1–5 (behaviour 1, recycling) are represented in Figs 8–12. SEM results for all 7 behaviours can be found in supplementary materials.

Second, we wanted to explore whether a moral component is already represented within the construct of attitudes or whether it is separate from attitudes (RQ2). Model 2 results

**Table 9. Goodness of fit results for the models subjected to SEM ($N$ = 181).**

| Fit index: | $X^2$ | $df$ | $X^2/df$ | P | CFI | RMSEA | SRMR | TLI |
|---|---|---|---|---|---|---|---|---|
| Goodness of fit criterion: | n/a | n/a | <3[a] | >0.05[b] | ≥0.95[c] | <0.05[d] | <0.08[c] | >0.80[b] |
| **Model 1: TPB without subjective norms** | | | | | | | | |
| 1.1. Recycling | 75.20 | 41 | 1.83 | 0.001 | 0.969 | 0.068 [0.043, 0.092] | 0.0376 | 0.958 |
| 1.2. Composting | 48.37 | 30 | 1.58 | 0.018 | 0.988 | 0.058 [0.024, 0.088] | 0.0429 | 0.982 |
| 1.3. Electron. Devices | 70.12 | 40 | 1.75 | 0.002 | 0.976 | 0.065 [0.038, 0.089] | 0.0468 | 0.967 |
| 1.4. Air Conditioning | 56.59 | 29 | 1.95 | 0.002 | 0.978 | 0.073 [0.044, 0.101] | 0.0550 | 0.966 |
| 1.5. Transport Use | 54.69 | 39 | 1.40 | 0.049 | 0.987 | 0.047 [0.003, 0.075] | 0.0427 | 0.982 |
| 1.6. Local Products | 64.61 | 30 | 2.15 | 0.000 | 0.969 | 0.080 [0.053, 0.107] | 0.0411 | 0.954 |
| 1.7. Plastic Bags | 71.89 | 41 | 1.75 | 0.002 | 0.975 | 0.065 [0.039, 0.089] | 0.0405 | 0.967 |
| **Model 2: TPB with moral norms as an additional predictor** | | | | | | | | |
| 2.1. Recycling | 72.46 | 48 | 1.50 | 0.013 | 0.978 | 0.053 [0.025, 0.077] | 0.0389 | 0.970 |
| 2.2. Composting | 74.39 | 38 | 1.96 | 0.000 | 0.977 | 0.073 [0.048, 0.097] | 0.0397 | 0.967 |
| 2.3. Electron. Devices | 74.48 | 48 | 1.56 | 0.008 | 0.982 | 0.056 [0.029, 0.079] | 0.0342 | 0.975 |
| 2.4. Air Conditioning | 133.96 | 71 | 1.89 | 0.000 | 0.962 | 0.070 [0.052, 0.088] | 0.0633 | 0.951 |
| 2.5. Transport Use | 135.44 | 71 | 1.91 | 0.000 | 0.963 | 0.071 [0.053, 0.089] | 0.0549 | 0.953 |
| 2.6. Local Products | 60.25 | 38 | 1.59 | 0.012 | 0.980 | 0.057 [0.027, 0.083] | 0.0420 | 0.971 |
| 2.7. Plastic Bags | 53.97 | 38 | 1.42 | 0.045 | 0.985 | 0.048 [0.008, 0.076] | 0.0352 | 0.979 |
| **Model 3: TPB with EPQ as an additional predictor** | | | | | | | | |
| 6.1. Recycling | 437.22 | 264 | 1.79 | 0.000 | 0.924 | 0.066 [0.057, 0.076] | 0.0603 | 0.914 |
| 6.2. Composting | 367.04 | 265 | 1.39 | 0.000 | 0.968 | 0.046 [0.034, 0.057] | 0.0565 | 0.963 |
| 6.3. Electron. Devices | 325.88 | 219 | 1.49 | 0.000 | 0.957 | 0.052 [0.040, 0.064] | 0.0527 | 0.950 |
| 6.4. Air Conditioning | 275.24 | 198 | 1.30 | 0.000 | 0.965 | 0.047 [0.032, 0.059] | 0.0637 | 0.959 |
| 6.5. Transport Use | 322.40 | 220 | 1.47 | 0.000 | 0.959 | 0.051 [0.038, 0.062] | 0.0538 | 0.952 |
| 6.6. Local Products | 348.68 | 220 | 1.58 | 0.000 | 0.946 | 0.057 [0.045, 0.068] | 0.0621 | 0.938 |
| 6.7. Plastic Bags | 432.08 | 264 | 1.64 | 0.000 | 0.935 | 0.059 [0.049, 0.069] | 0.0613 | 0.926 |
| **Model 4: TPB with moral norms as a predictor of attitudes** | | | | | | | | |
| 3.1. Recycling | 109.66 | 50 | 2.19 | 0.000 | 0.947 | 0.081 [0.061, 0.102] | 0.0750 | 0.929 |
| 3.2. Composting | 84.90 | 31 | 2.34 | 0.000 | 0.962 | 0.098 [0.074, 0.124] | 0.0638 | 0.944 |
| 3.3. Electron. Devices | 102.88 | 50 | 2.06 | 0.000 | 0.965 | 0.077 [0.055, 0.098] | 0.0544 | 0.953 |
| 3.4. Air Conditioning | 160.78 | 73 | 2.20 | 0.000 | 0.947 | 0.082 [0.065, 0.099] | 0.0761 | 0.934 |
| 3.5. Transport Use | 186.20 | 73 | 2.55 | 0.000 | 0.936 | 0.062 [0.076, 0.109] | 0.0770 | 0.920 |
| 3.6. Local Products | 67.46 | 40 | 1.69 | 0.004 | 0.974 | 0.062 [0.035, 0.087] | 0.0502 | 0.966 |
| 3.7. Plastic Bags | 81.40 | 40 | 2.04 | 0.000 | 0.962 | 0.076 [0.052, 0.099] | 0.0663 | 0.948 |
| **Model 5: TPB with EPQ as a predictor of attitudes** | | | | | | | | |
| 7.1. Recycling | 495.95 | 267 | 1.86 | 0.000 | 0.918 | 0.069 [0.059, 0.078] | 0.0678 | 0.907 |
| 7.2. Composting | 371.62 | 267 | 1.29 | 0.000 | 0.967 | 0.047 [0.035, 0.058] | 0.0620 | 0.963 |
| 7.3. Electron. Devices | 329.24 | 221 | 1.49 | 0.000 | 0.903 | 0.070 [0.062, 0.079] | 0.0816 | 0.892 |
| 7.4. Air Conditioning | 282.92 | 201 | 1.41 | 0.000 | 0.963 | 0.048 [0.034, 0.060] | 0.0602 | 0.958 |
| 7.5. Transport Use | 333.46 | 223 | 1.50 | 0.000 | 0.955 | 0.052 [0.040, 0.064] | 0.0644 | 0.949 |
| 7.6. Local Products | 348.81 | 222 | 1.57 | 0.000 | 0.947 | 0.056 [0.045, 0.067] | 0.0622 | 0.940 |
| 7.7. Plastic Bags | 433.73 | 266 | 1.63 | 0.000 | 0.935 | 0.059 [0.049, 0.069] | 0.0637 | 0.926 |

CFI, Comparative Fit Index; RMSEA, Root Mean Square Error of Approximation [95% CI]; SRMR, Standardised Root Mean Square Residual; TLI, Tucker–Lewis Index.

aValues recommended by van Dam [81].

bValues recommended by Hu and Bentler [78].

cValues recommended by Hooper et al. [74].

dValues recommended by MacCallum et al. [75].

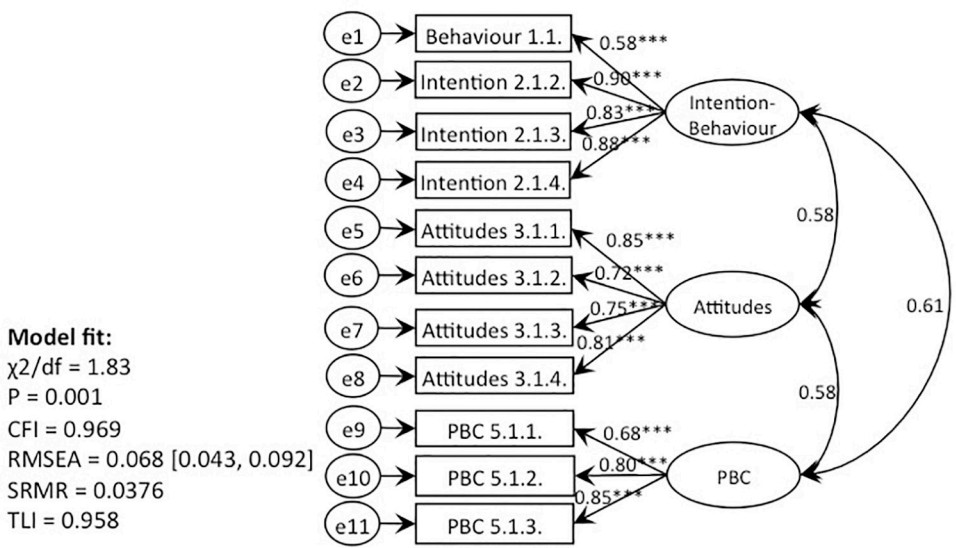

**Fig 4. CFA, behaviour 1 (recycling): TPB without subjective norms (Model 1).**

showcased that moral norms significantly predicted the 'Intention-Behaviour' of all behaviours except local product purchasing. However, the inspection of Model 3 results revealed that the addition of relativism and idealism provided the opposite pattern. Specifically, idealism did not predict any of the 'Intentions-Behaviours' significantly. Similarly, relativism significantly predicted only one of the 'Intention-Behaviour' constructs (composting, P < 0.05). Furthermore, the addition of moral norms (Model 2) decreased the predictive capacity of attitudes to the non-significant level for behaviours 2–8 (as compared to Model 1) and has reduced their predictive capacity from P < 0.001 to P < 0.05 for behaviour one (recycling) and nine (plastic bags). In contrast, attitudes remained a significant predictor when EPQ was added to the TPB

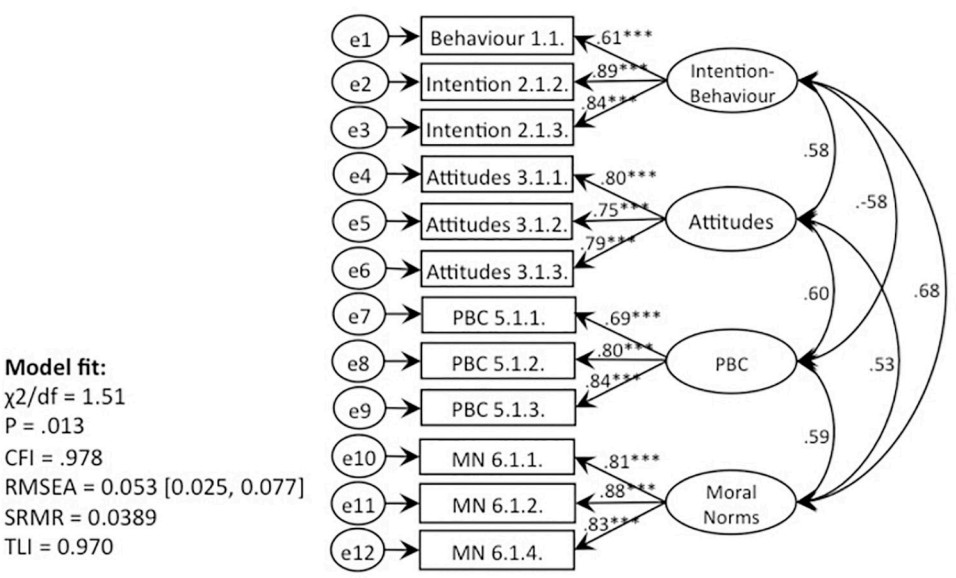

**Fig 5. CFA, behaviour 1 (recycling): TPB with moral norms (Models 2, 4).**

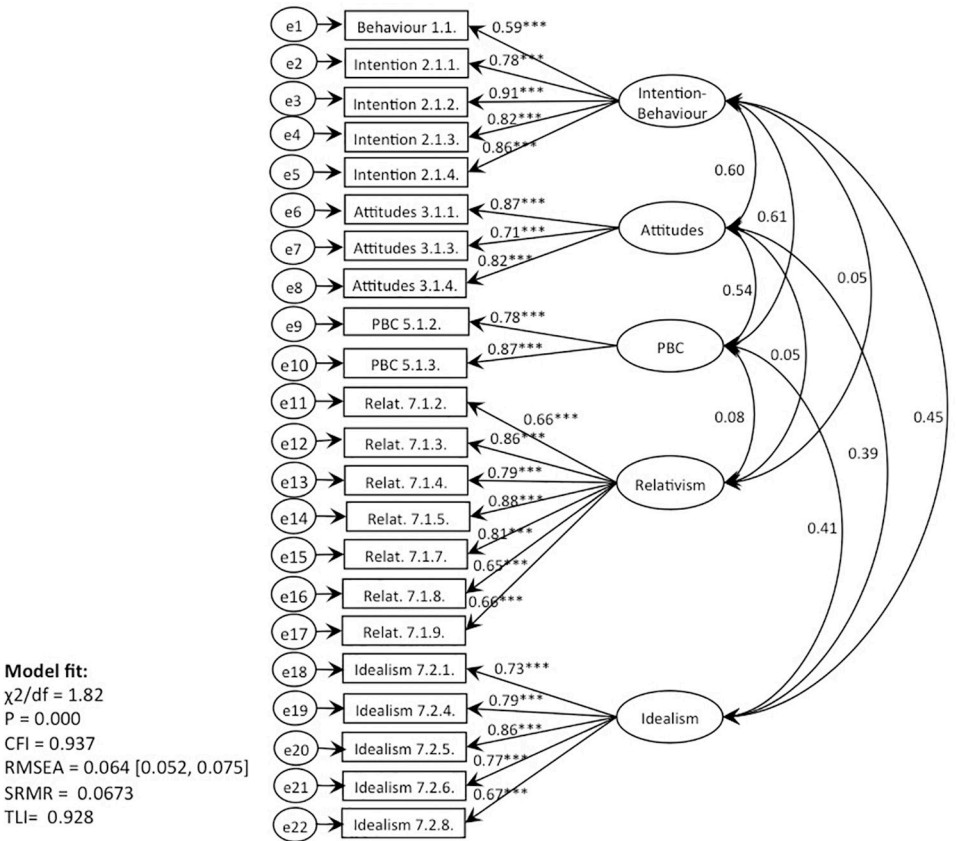

**Fig 6. CFA, behaviour 1 (recycling): TPB with EPQ (Models 3, 5).**

framework (Model 3 compared to Model 1). This could suggest that moral norms represent the construct of attitudes, but EPQ does not. Furthermore, positioning moral norms as a predictor of attitudes (Model 4) revealed that both moral norms and attitudes remained significant predictors, supporting the statement above that moral norms are represented within the attitudes. However, positioning EPQ as a predictor of attitudes revealed a different pattern (Model 5). Specifically, relativism did not predict any of the 'attitude' constructs significantly, except for local product purchasing (behaviour 7). In contrast, idealism significantly predicted all the 'attitude' constructs. Notably, the attitudes also remained a significant predictor, suggesting that the 'attitude' construct has an idealistic component that, in turn, impacts the 'Intention-Behaviour' variable. However, the relativistic component was not found within 'attitudes' nor within 'Intention-Behaviour' constructs, suggesting that relativistic philosophy is unrelated to the pro-environmental domain (Tables 10 and 11).

Third, further inspection of whether moral norms (Model 4) predicted attitudes in a stronger way than EPQ (Model 5) revealed that $R^2$ of attitudes was higher when moral norms rather than EPQ were added to the TPB framework (RQ3; Fig 13). This suggests that moral norms are a better-fitting moral element within the TPB framework.

Fourth, Models 2, 4, and 5 revealed that all behaviours were significantly predicted by the moral element. Even though the addition of EPQ to Model 3 did not result in EPQ significantly predicting 'Intentions-Behaviours' (except for behaviour 2, composting), the overarching direction suggests that this might be due to the incorrect placement of EPQ within the TPB

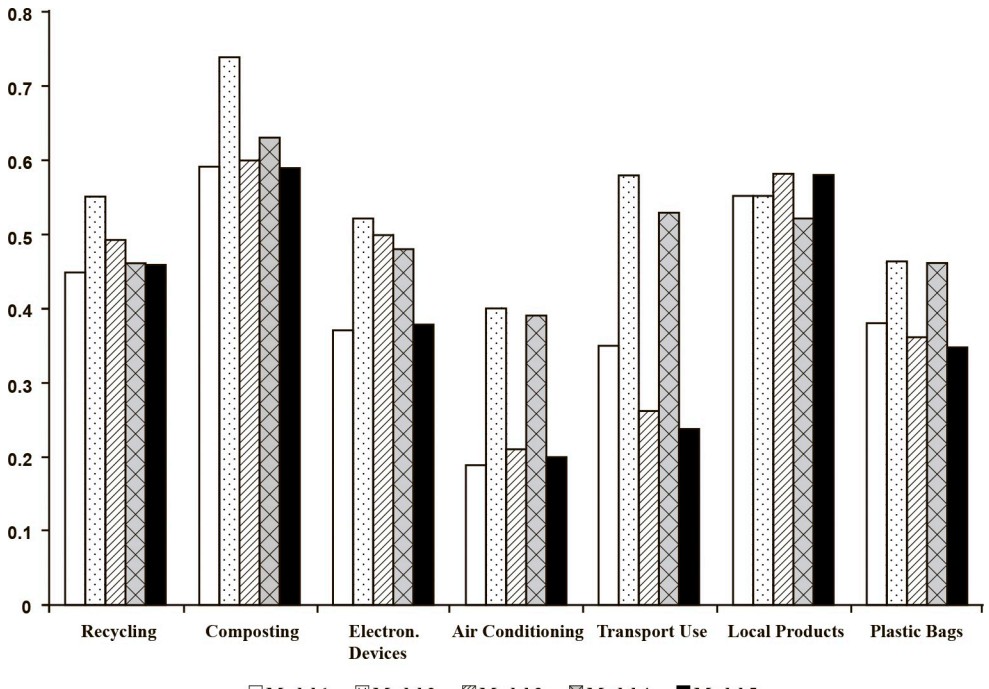

**Fig 7. R² changes in relation to the addition and placement of the moral element (N = 181).**

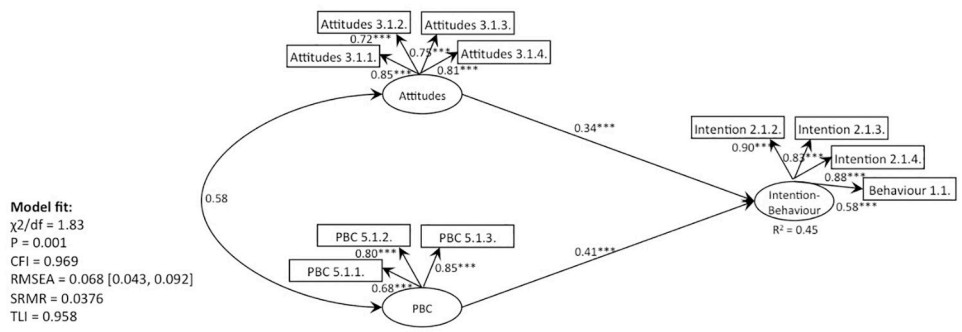

**Fig 8. SEM, behaviour 1 (recycling): TPB without subjective norms (Model 1).**

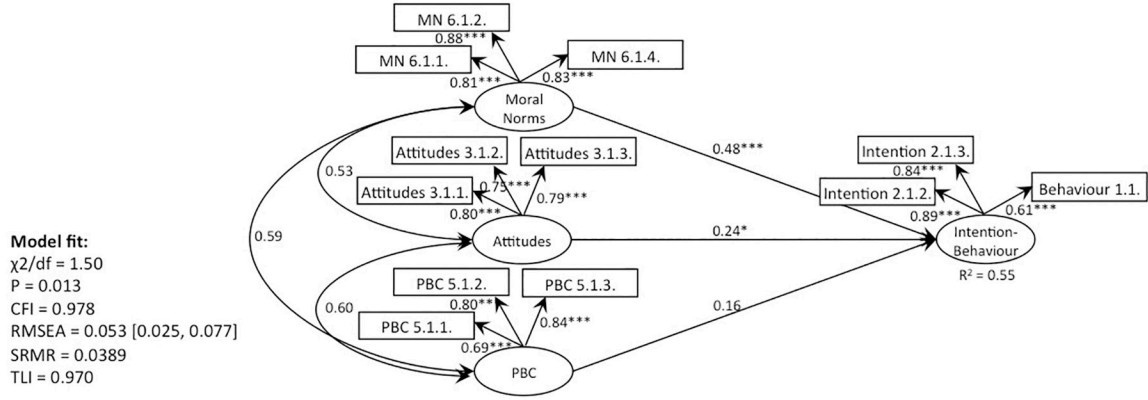

**Fig 9. SEM, behaviour 1 (recycling): TPB with moral norms as additional predictor (Model 2).**

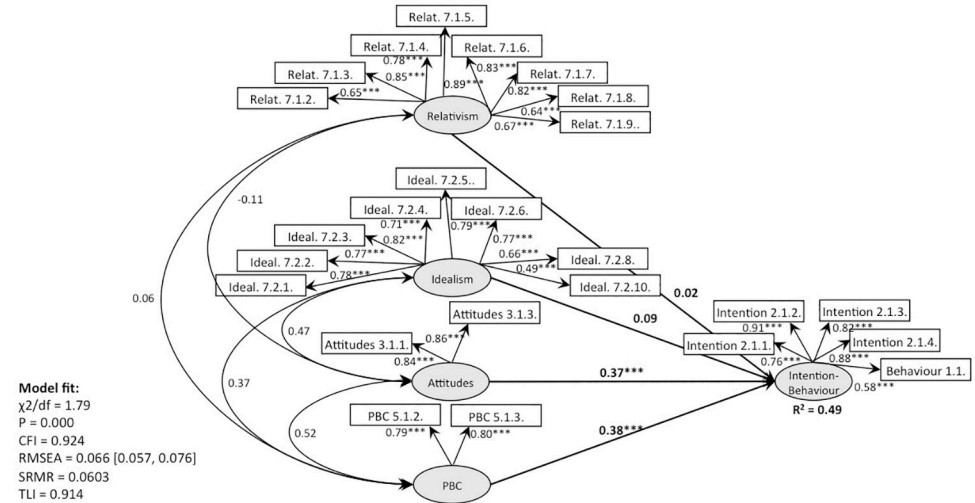

**Fig 10. SEM, behaviour 1 (recycling): TPB with EPQ as additional predictor (Model 3).**

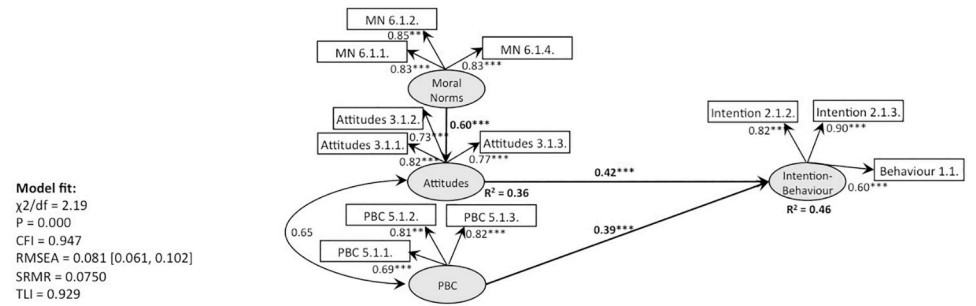

**Fig 11. SEM, behaviour 1 (recycling): TPB with moral norms as predictor of attitudes (Model 4).**

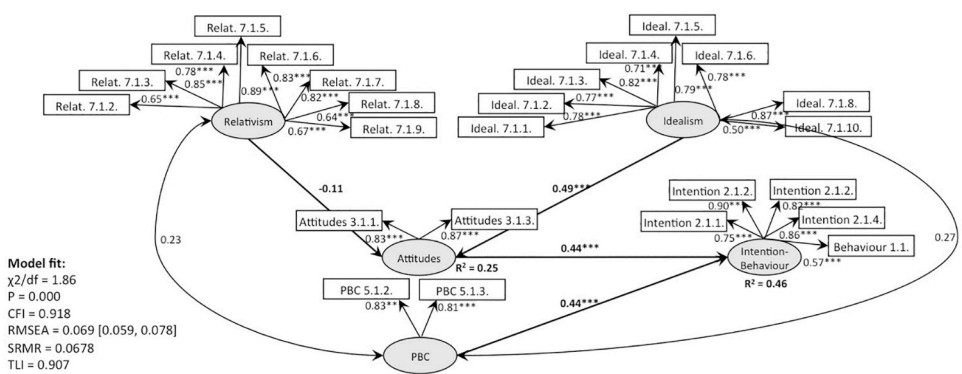

**Fig 12. SEM, behaviour 1 (recycling): TPB with EPQ as predictor of attitudes (Model 5).**

**Table 10. Path analyses—TPB ($N = 181$).**

| Behaviours | | Attitudes | PBC |
|---|---|---|---|
| Model 1 (TPB without subjective norms) | | | |
| 1. Recycling | | 0.34*** | 0.41*** |
| 2.Composting | | 0.26** | 0.59*** |
| 3.El. Devices | | 0.56*** | 0.08 |
| 4.Air Cond. | | 0.40*** | 0.09 |
| 5.Transp. Use | | 0.44*** | 0.25** |
| 7.Local Prod. | | 0.52*** | 0.33*** |
| 9.Plastic Bags | | 0.52*** | 0.17* |

framework. Hence, it appears that all seven behaviours were driven by morality (RQ4, Tables 10 and 11 and Figs 8–12).

Fifth, the question of whether the 'attitude-behaviour' gap occurs between attitudes and intentions or intentions and behaviours is the most difficult to answer (RQ5). This is because we were not able to separate the construct of 'intentions' from the construct of 'behaviours,' which subsequently suggests that 'intentions' and 'behaviours' is the same variable. Furthermore, the findings revealed that the moral element significantly predicted the 'Intention-Behaviour' variable (Models 2, 4, 5), suggesting that there's no 'attitude-behaviour' gap. This contradicts the findings of Zaikauskaite and colleagues [15], who used morality-centered Hunt-Vitell's framework to predict the same behaviours. Hence, it's unclear whether the addition of either moral norms or EPQ did not inflate the effects of morality within the TPB framework (Tables 10 and 11 and Figs 8–12).

## General discussion

The present study was designed to investigate the role of morality in predicting pro-environmental behaviours within the TPB framework. Specifically, we aimed to explore (i) whether the addition of a moral component to the TPB model increases its predictive power; (ii) whether the TPB construct of attitudes already represents a moral component within it, or whether the constructs of attitudes and morality complement the prediction of outcome variables separately; (iii) whether EPQ is a more suitable measure of morality within the TPB framework, as compared to the measure of moral norms; (iv) whether different pro-environmental behaviours are equally driven by the moral element, or whether some behaviours have no relationship with moral dimension; (v) whether it is a moral dimension that drives 'attitude-behaviour' gap, and, if so, whether the gap occurs between attitudes and intentions, or intentions and behaviours. In doing so, we have tested the effects of a moral component on ten everyday pro-environmental behaviours, such as recycling, composting, purchasing energy-efficient appliances, etc. [60]. To the best of our knowledge, no study to date has investigated the effectiveness of moral norms vs. EPQ within the TPB framework using ten pro-environmental behaviours. Therefore, our study serves as the first example addressing the aforementioned points.

First of all, we found that the addition of the moral element to the TPB framework has increased the predictive capacity for 23 out of 28 TPB models, although the reasons why 5 of the models resulted in a slight decrease of the $R^2$ are not clear. Indeed, having a relatively small sample size makes it difficult to say whether the results of the five models are not misrepresented and are specific to how morality predicts transport and plastic bag use and the purchasing of local products. Former studies by Zaikauskaite and colleagues [15, 58], who investigated

**Table 11. Path analyses–MN and EPQ ($N = 181$).**

| | Model 2 (MN as Additional Predictor) | | | Model 3 (EPQ as Additional Predictor) | | | |
| --- | --- | --- | --- | --- | --- | --- | --- |
| Behaviours | Moral Norms | Attitudes | PBC | Relativism | Idealism | Attitudes | PBC |
| 1.Recycling | 0.48*** | 0.24* | 0.18 | 0.02 | 0.09 | 0.36*** | 0.38*** |
| 2.Composting | 0.66*** | -0.13 | 0.42*** | 0.02* | 0.12 | 0.26** | 0.56 |
| 3.El. Devices | 0.62*** | 0.04 | 0.14 | -0.06 | 0.12 | 0.50*** | 0.10 |
| 4.Air Cond. | 0.52*** | 0.11 | 0.08 | -0.04 | 0.09 | 0.40*** | 0.05 |
| 5.Transp. Use | 0.66*** | 0.01 | 0.19* | 0.08 | 0.14 | 0.20* | 0.32*** |
| 7.Local Prod. | -0.18 | 0.82 | 0.11 | 0.02 | -0.01 | 0.53*** | 0.33*** |
| 9.Plastic Bags | 0.44*** | 0.22* | 0.18* | 0.08 | 0.04 | 0.46*** | 0.17 |
| | Model 4 (MN as Predictor of Attitudes) | | | Model 5 (EPQ as Predictor of Attitudes) | | | |
| Behaviours | Moral Norms | Attitudes | PBC | Relativism | Idealism | Attitudes | PBC |
| 1.Recycling | 0.60*** | 0.42*** | 0.39*** | -0.11 | 0.49*** | 0.44*** | 0.44*** |
| 2.Composting | 0.64*** | 0.20** | 0.68*** | 0.09 | 0.43*** | 0.25** | 0.59*** |
| 3.El. Devices | 0.86*** | 0.83*** | 0.15 | 0.08 | 0.44*** | 0.53*** | 0.15 |
| | | | $p = 0.52$ | | | | |
| 4.Air Cond. | 0.81*** | 0.61*** | 0.06 | 0.01 | 0.39*** | 0.42*** | 0.10 |
| 5.Transp. Use | 0.82*** | 0.63*** | 0.21** | 0.03 | 0.45*** | 0.30** | 0.37 |
| 7.Local Prod. | 0.85*** | 0.55*** | 0.26*** | 0.19* | 0.35*** | 0.54*** | 0.33*** |
| 9.Plastic Bags | 0.74*** | 0.61*** | 0.18* | 0.09 | 0.45*** | 0.51*** | 0.14 |

the effects of morality on the same ten behaviours, did not find any differences in how the moral component predicts different pro-environmental behaviours. Notably, these studies were based on regression analyses [58] and Hunt-Vitell's framework of the General Theory of Marketing Ethics, which assumes morality as a central rather than an additional parameter [15]. Hence, the question of whether obtained findings relating to the five models is not misrepresented due to sample size or other factors or whether they are unique to the TPB framework calls for further investigation. Some of the previous studies found that moral norms had increased $R^2$ or the model fit of the TPB [33, 39], whereas others found the opposite even in cases when moral norms contributed to predicting outcome variable(s) significantly [39], the decrease of $R^2$; or Kaiser and Scheuthle [26], Yazdanpanah and Forouzani [40]: the decrease of the model fit).

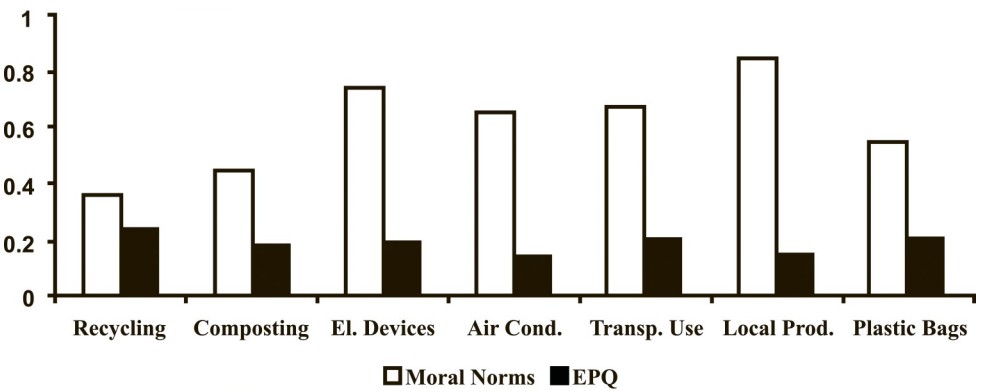

**Fig 13. $R^2$ changes in the construct of 'attitudes' with moral norms vs. EPQ as predictors ($N = 181$).**

In contrast, some researchers reported that the inclusion of moral norms has merely contributed to increasing the predictive power of the TPB framework [26, 37]. In this vein, our results support the direction that the addition of a moral element (either moral norms or EPQ) increases $R^2$ of the TPB framework, although CFI fit was slightly poorer for models with EPQ, and RMSEA fit was slightly poorer for models with moral norms. Further research is necessary to reveal whether the changes in fit indices are meaningful.

Second, we found that adding moral norms as an additional predictor of 'Intention-Behaviour' significantly predicted 'Intention-Behaviour' of all of the behaviours except local product purchasing, whereas adding EPQ did not result in significant 'Intention-Behaviour' predictions (except relativism significantly predicting composting, $P < 0.05$). Whether the exceptions in the cases of local product purchasing and composting are meaningful or whether the results are misrepresented due to small sample size (etc.) is a question of further research. However, the general direction reveals that adding moral norms nullifies the predictive power of attitudes, whereas adding EPQ neither changes the predictive power of attitudes nor serves as a significant predictor itself. This is an interesting finding because it suggests that moral norms are already reflected in the construct of attitudes. Indeed, a further investigation of placing moral norms as a predictor of attitudes rather than an additional predictor revealed that in this situation, moral norms significantly predicted attitudes and attitudes significantly predicted 'Intentions-Behaviours.' These findings are in line with a study by Chan and Bishop [24], who found that moral norms were highly correlated with attitudes ($r = 0.69$), suggesting that the two constructs possibly overlap each other and thus may lack convergent validity. As a result, the authors replaced attitudes with moral norms and found that this model fits the data extremely well. In a similar vein, our aggregate correlation results revealed that moral norms were also highly correlated with attitudes ($r = 0.70$, $P < 0.001$), thus calling for further research on how to use the constructs of attitudes and moral norms within the TPB framework more effectively (i.e., is it necessary to use both of the constructs, etc.).

Notably, the case with the addition of another moral element, EPQ, was different. The aggregate correlation coefficient of attitudes was much lower for idealism ($r = 0.48$, $P < 0.001$) and further reduced by half for relativism ($r = 0.24$, $P < 0.01$), as compared to the correlation coefficient of attitudes and moral norms ($r = 0.70$, $P < 0.001$). Thus, the results withhold the possibility that the constructs of EPQ and attitudes lack convergent validity. However, the inclusion of EPQ as an additional predictor did not predict 'Intentions-Behaviours' significantly (except in a case of relativism predicting composting, $\beta = 0.02$, $P < 0.05$), which was in contrast to the case of moral norms. The finding that relativism does not predict pro-environmental behaviours has already been showcased by the study of Zaikauskaite and colleagues [58]. However, the reasons why idealism did not predict 'Intentions-Behaviours' was less clear. Further placement of EPQ as a predictor of attitudes revealed that idealism significantly predicted attitudes of all the seven behaviours. In contrast, relativism did not (except local products purchasing, $\beta = 0.19$, $P < 0.05$), suggesting that attitudes already represent a certain degree of idealism within them. Further investigation of the aggregate correlation between moral norms and idealism showcases that the two constructs are relatively highly correlated ($r = 0.60$, $P < 0.001$). Thus, it's possible that the construct of moral norms already represents an idealistic component within them. For this reason, both moral norms and idealism predicted attitudes in a similar (significant) manner.

Third, these findings pose a further question of whether EPQ is a more suitable measure of morality within the TPB framework, as compared to the measure of moral norms. Indeed, the value of adding relativism to the TPB framework is the most obscure. That is, if relativism does not predict attitudes or behaviours, then perhaps amending it altogether would be the most effective option. However, the question of whether moral norms serve as a more accurate

predictor of morality than idealism is more difficult. This is because it's unclear whether one measure is more immune to social desirability bias than the other and also whether measuring different aspects of morality (i.e., norms vs. idealistic component) impacts the prediction accuracy in a different way. For example, it's unclear what the finding that moral norms predict 'Intentions-Behaviours' when included as an additional predictor, but idealism does not, means in a broader perspective. One of the avenues of interpretation could suggest that moral norms include a higher component of social desirability bias. Thus the way they predict 'Intentions-Behaviours' is inflated. Indeed, it may be that idealism incorporates a lesser degree of social desirability bias and, therefore, does not result in predicting 'Intentions-Behaviours' when put as an additional predictor. Hence, further research on how accurately the two constructs reflect unbiased moral elements could shed light on their effectiveness in indicating real-life rather than theoretical actions.

Fourth, we have found that all behaviours were equally impacted by morality. That is, we have found a general direction that both moral norms and idealism impact attitudes towards the tested pro-environmental behaviours, which is in line with the previous research assessing the impact of a moral component on the same behaviours [15, 58]. Nevertheless, it's worth noting that the reported results represent information acquired on the conscious level. The fact that participants rationally indicated a certain degree of morality does not necessarily mean that the level is not overestimated due to the need to think of and/or present oneself as a socially responsible person. As suggested by Markowitz and Shariff [44], climate change issues lack intuitive moral judgment, meaning that the lack of implicit moral associations may be the reason why rational choice behaviour frameworks such as the TPB overestimate the effects of morality and thus over represent its impact in real-life situations.

Fifth, the overrepresentation of the impact of morality may be the reason why our results did not indicate the presence of the 'attitude-behaviour' gap. Although we expected to find an answer to whether the gap occurs between attitudes and intentions, or intentions and behaviours, our findings suggested that there is no statistical difference between the constructs of intentions and behaviours, leading to a conclusion that there is no gap between the two. However, we did not find a gap between attitudes and intentions either. This is an unexpected finding because both the regression-based study by Zaikauskaite and colleagues [58] and Zaikauskaite et al.'s study [15] using Hunt-Vitell's framework incorporated the same pro-environmental behaviours and have indicated the presence of 'attitude-behaviour' gaps, although the specific point at which the gaps occurred was less clear.

## Limitations

Admittedly, the present study is limited in some key respects. Foremost among these is the notion that we have measured self-reported rather than actual pro-environmental behaviours. Vezich and colleagues [82] examined the real-life representation of such a case by correlating self-reports with neural activity in order to examine whether activity in the specific neural structures corresponded to the results of the self-reported pro-environmental behaviour measures. Specifically, the authors proposed that consumers' willingness to purchase sustainable products depends on competing value signals, and these signals may be misidentified in self-reports. Indeed, the authors found fMRI support for the idea that neuronal activity relating to viewing 'green' advertisements differs from the self-reported liking of those advertisements [82]. The findings revealed that self-reported liking correlated with neuronal activity for control but not for the 'green' ads, thus providing evidence for the presence of bias. Hence, it's unclear how closely the findings reflect pro-environmental behaviour tendencies of real life.

Furthermore, it's unclear how much the scale measuring social norms is prone to measuring social desirability bias. Indeed, pro-environmental behaviours are highly desirable, and social norms measure one's adherence to desirable social norms. Tajfel's theory of social identity and intergroup behaviour [83] suggests that people tend to maintain desirable social identity in the eyes of others because evidence of behaving undesirably may contribute to the social identity threat [84, 85]. In turn, social identity threat may lead to perceiving one as an out-group rather than an in-group member, and this status change, alongside the individual's rejection from the group, can cause a fine level of inconvenience to the individual's everyday life. Hence, it's difficult to tease out the measurement of social norms from measuring social desirability rather than the actual level of an individual's adherence to social norms.

For this reason, we have separated the survey into the six different studies believing that this approach might help reduce socially desirable responses. We believed this approach could be useful because the first survey measured one's adherence to ten environmental behaviours (e.g., recycling, composting, etc.). Hence, we believed that not telling the participant about our intent to invite them to participate in the further surveys and leaving one month apart from the first and second surveys would be long enough for the participant to forget both our questionnaire items and their response to those. In addition, we have split the rest of the survey into five parts because the measurement scales are quite repetitive, and we did not want to cause boredom to the participants. Hence, we preferred to provide five shorter questionnaires over the next upcoming days rather than one long questionnaire that the participant would be required to complete in one go. Such an approach has been previously suggested by Zaikauskaite and colleagues [58], who have studied the relationship between morality and pro-environmental behaviours using the same ten pro-environmental behaviour items [60], and utilised by Zaikauskaite and colleagues [15] who have also studied the relationship between morality and pro-environmental behaviours using the same ten pro-environmental behaviour items [60]. In the latter study, the questionnaire was split into two parts where the survey measuring pro-environmental behaviour was launched one month prior to the rest of the survey, measuring the effectiveness of moral dimension using the GTME framework [3, 4]. Interestingly, the study's results have demonstrated that GTME framework revealed the presence of the 'attitude-behaviour' gap. Hence, we attempted to replicate these findings using the TPB model, the same pro-environmental behaviour measurements [60], and a methodological approach of separating pro-environmental behaviours from the rest of the survey items. Indeed, we have underestimated the efficiency of such an approach in terms of the difficulty of receiving six completed parts of the survey coming from the same participant. Yet, we do not have the ground to believe that such an approach somehow undermined the quality of the final responses because participants made no obligations to complete all six parts of the survey (as intended, they did not know that six questionnaire parts existed).

Next, each of the pro-environmental behaviour was measured using one item. However, the rest of the constructs (except EPQ) were measured using four items per behaviour, which might have been the reason why the constructs of intentions and behaviours were statistically indistinct. An alternative behaviour measurement that provides more variability could help separate the constructs into two factors.

Last, it should be indicated that the sample size of this study was relatively small due to a high volume of participants not passing attention checks. Notably, MTurk participants are anonymous, complete studies in unsupervised settings and unknown locations, and are motivated by financial incentives [86]. Hence, it's possible that a larger sample size would have reduced the potential fluctuation in the averages of the measurements and thus provided more accurate results. Nevertheless, the overall directionality of acquired results is in line with reviewed literature.

## Implications & future directions

Our study provides several theoretical implications. First, our results have demonstrated that the TPB model may deliver inaccurate findings when the questionnaire is composed of the self-reported measurement items. Here, our results suggested no 'attitude-behaviour' gap between attitudinal and behavioural variables. However, such a result is hardly accurate because academic and industry cases note the 'attitude-behaviour' gap as the main reason why pro-environmental attitudes do not translate into pro-environmental behaviours. Hence, our findings demonstrate how rigorous assessment using the TPB framework can actually provide inaccurate results. Second, we have demonstrated that the addition of the moral component did not help reveal the 'attitude-behaviour' gap. Contrary to the previous findings [15, 58], the moral element has been identified as the variable to explain the 'attitude-behaviour' gap. Specifically, the studies [15, 58] have found that morality is not fully integrated into pro-environmental behaviours, and environmental attitudes do not translate to environmental behaviours because it's morality that does not translate from attitudes to behaviours. Hence, the attempt to increase the prediction of the 'attitude-behaviour' gap using two different elements of morality (i.e., moral norms and EPQ) strengthens the findings that our study has delivered inaccurate results because it failed to reveal the presence of the 'attitude-behaviour' gap even when the TPB model was enhanced with a moral element. This is an important implication that raises the awareness of why theory may not reflect real-life results, as well as invites to beware of the flawed interpretations of the findings even in cases providing adequate model fit and significant findings. Hence, future studies assessing pro-environmental behaviours should carefully consider the accuracy of the acquired results.

From a methodological perspective, our study demonstrates the inefficiency of dividing the questionnaire into six different parts and launching them separately on MTurk. Notably, the MTurk platform does not offer an opportunity to recruit the same participants for the separate studies and provide compensation after the completion of all the studies. Hence, we have used the participant's IDs to invite them to every following study that we wanted to acquire their responses for. However, whether or not the participants wanted to participate further was a completely voluntary choice. Clearly, we did not expect such a low participation rate with regard to completing all six parts of the questionnaire.

Nevertheless, this does not mean that the responses of the participants who completed the questionnaire were more valid or invalid. Simply, a low participation rate meant that the participants hadn't chosen to complete the next study, and had no obligations to do so. On the one hand, it would have been more efficient if MTurk offered an opportunity to recruit the participants for the desired number of studies and provide compensation after the completion of the last one, ensuring a much higher participation rate. On the other hand, we aimed to minimise social desirability bias by not telling the participants that there would be the next study to complete. Hence, it's unclear whether we have actually had any success in minimising the bias and whether the results would differ if a full questionnaire were launched at once. Future studies could attempt to investigate whether dividing the questionnaire into six parts helps to minimise the bias by launching a study consisting of six parts and launching the same study without dividing it into six parts. Indeed, knowing whether there is a difference in the obtained response patterns would help understand whether it's worth dividing the study regardless of the inefficiency such an approach might cause.

## Conclusion

This study extends the research on the effects of morality in a pro-environmental domain. The findings have demonstrated that the addition of the moral element to the TPB framework did

not reveal the presence of the 'attitude-behaviour' gap, despite both moral norms and idealism significantly predicting pro-environmental attitudes. The findings do not provide an indication of whether moral norms or idealism should be used as a more accurate measure of morality within the TPB framework, although relativism was found to have no significant effects. Further investigation of why the moral element does not reveal the 'attitude-behaviour' gap within the TPB framework predicting pro-environmental behaviours would help understand the reasons why rational choice models tend to overestimate theoretical vs. real-life engagement with sustainability.

## Supporting information

**S1 Fig. CFA, original TPB (Model 1).**
(PDF)

**S2 Fig. CFA, TPB without subjective norms (adjusted Model 1).**
(PDF)

**S3 Fig. CFA, TPB with moral norms (adjusted Models 2, 4).**
(PDF)

**S4 Fig. CFA, TPB with EPQ (adjusted Models 3, 5).**
(PDF)

**S5 Fig. SEM, original TPB (Model 1).**
(PDF)

**S6 Fig. SEM, TPB without subjective norms (adjusted Model 1).**
(PDF)

**S7 Fig. SEM, TPB with moral norms as additional predictor (adjusted Model 2).**
(PDF)

**S8 Fig. SEM, TPB with EPQ as additional predictor (adjusted Model 3).**
(PDF)

**S9 Fig. SEM, TPB with moral norms as predictor of attitudes (adjusted Model 4).**
(PDF)

**S10 Fig. SEM, TPB with EPQ as predictor of attitudes (adjusted Model 5).**
(PDF)

**S1 Table. Shapiro-Wilk tests for normality assumptions.**
(PDF)

**S2 Table. EFA, original TPB (Model 1).**
(PDF)

**S3 Table. EFA, TPB without subjective norms (adjusted Model 1).**
(PDF)

**S4 Table. EFA, TPB with moral norms (adjusted Models 2, 4).**
(PDF)

**S5 Table. EFA, TPB with EPQ (adjusted Models 3, 5).**
(PDF)

**S1 Text. Experimental protocol.**
(PDF)

## Acknowledgments

We would like to thank undergraduate students Farhana Begum, Desiree Cho, Yuliya Dubinska, Xinyu Jiang, Betsy H. L. Kwok, Yifan Liang, Hannah E. Mildner, Nicole S. Ngwenya, Klara E.J. Selén, Martyna K. Stasiak, Evie J. Pyper, Clara L. Velasco, Chloe A. Yu for their work on the questionnaire items and informal data analysis. We would like to express our gratitude to Jada-Rose Dixon for invaluable assistance with this research piece. We would like to thank the Department of Experimental Psychology (Undergraduate Student Labs) for funding this research.

## Author Contributions

**Conceptualization:** Laura Zaikauskaitė.

**Data curation:** Laura Zaikauskaitė.

**Formal analysis:** Laura Zaikauskaitė, Alicja Grzybek, Rachel E. Mumford.

**Funding acquisition:** Laura Zaikauskaitė.

**Investigation:** Laura Zaikauskaitė, Alicja Grzybek, Rachel E. Mumford.

**Methodology:** Laura Zaikauskaitė.

**Project administration:** Laura Zaikauskaitė.

**Supervision:** Dimitrios Tsivrikos.

**Validation:** Laura Zaikauskaitė, Alicja Grzybek, Rachel E. Mumford, Dimitrios Tsivrikos.

**Visualization:** Laura Zaikauskaitė, Alicja Grzybek, Rachel E. Mumford.

**Writing – original draft:** Laura Zaikauskaitė.

**Writing – review & editing:** Laura Zaikauskaitė, Alicja Grzybek, Rachel E. Mumford, Dimitrios Tsivrikos.

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
