## [Decision Letter · Decision Letter 0]

12 May 2023

PONE-D-22-33487The Theory of Planned Behaviour Doesn't Reveal 'Attitude-Behaviour' Gap? Contrasting the Effects of Moral Norms vs. Idealism and Relativism in Predicting Pro-Environmental BehavioursPLOS ONE

Dear Dr. Zaikauskaite,

Thank you for submitting your manuscript to PLOS ONE. After careful consideration, we feel that it has merit but does not fully meet PLOS ONE’s publication criteria as it currently stands. Therefore, we invite you to submit a revised version of the manuscript that addresses the points raised during the review process.

ACADEMIC EDITOR:Thank you for your submission to PLOS ONE. Two reviewers have now commented on your manuscript. As you can see from their reviews, both find the study to be interesting and to address relevant questions. At the same time, they both have several suggestions for improvement. I have therefore decided to invite you to submit a revised version of the manuscript addressing the reviewers' concerns.

We look forward to receiving your revised manuscript.

Kind regards,

Nicola Diviani

Academic Editor

PLOS ONE

Journal Requirements:

Reviewers' comments:

Reviewer's Responses to Questions

**Comments to the Author**

1. Is the manuscript technically sound, and do the data support the conclusions?

Reviewer #1: Yes

Reviewer #2: Partly

2. Has the statistical analysis been performed appropriately and rigorously? 

Reviewer #1: Yes

Reviewer #2: No

3. Have the authors made all data underlying the findings in their manuscript fully available?

Reviewer #1: Yes

Reviewer #2: Yes

4. Is the manuscript presented in an intelligible fashion and written in standard English?

Reviewer #1: Yes

Reviewer #2: Yes

5. Review Comments to the Author

Reviewer #1: The authors of this study developed a survey based on several self-reported measures related to pro-environmental attitudes, alongside moral norms and confidence in the ability to pursue such behaviours (perceived behavioural control), to test their efficacy in predicting pro-environmental behaviours and/or intentions. To this purpose, they used complex statistical analyses, including path analysis/structural equation modeling, to test 5 models for each of several enviromental behaviours. Among many others, a key question of the study is whether the so-called "attitude-behaviour gap", i.e., the inconsistency between pro-environmental attitudes and behaviours, is driven (and can thus be explained) by the presence of moral attitudes that do not translate into consistent moral (pro-environmental) behaviours. Data from 181 retained participants of an online-survey (out of 431 participating) showed that, while pro-environmental attitudes were predicted both by moral norms and the "idealism" scale of the Ethics Position Questionnaire (EPQ), the inclusion of these metrics did not reveal an "attitude-behaviour gap". All in all, these data provide partial answers to interesting questions on relevant topics. In my view, this is an interesting and well-performed study, that however would require more effort to help readers appreciate some important conclusions.

1) Although this is already briefly discussed in a "Limitation" section, the authors should discuss in more depth the trade-off between the complexity of their statistical models and the quality of exclusively self-reported data collected in online surveys with 62% of rejected participants. I wonder whether, at this point, it would be worth collecting data based on "real" behaviours in a more ecological setting, that might reasonably result in a sample size not much smaller than 181.

2) It is clear that there might be large overlap, in terms of the measured psychological construct, among the several questionnaires included in this study. I would then suggest to state explicitly that preliminary statistical analyses allowed to take into account such overlap. In other words, I wonder to what extent their analytic methods allow to assess whether these scales measure distinctive vs. overlapping constructs.

3) I wonder whether, and to what extent, the "subject norms" scale is actually measuring something related to social desiderability. Can the authors express their opinion in this regard?

4). p.16: can the authors explain why their procedure is expected to minimize a "social desiderability" bias?

5) only 181 out of 431 participanst were retained in stats: this is a huge proportion, that should be at least commentd/discussed by the authors to reassure readers about the reliability of the procedure. At least, can we expect compliance by the 38% of retained participants...? What is the added value of collecting self-reported variables from "large" sample, compared with well-controlled experiments measuring "real" (although maybe simulated) behaviours from "smaller" samples?

6) Figure 5: to help the reader, it would be useful to recap here the factors included in the model.

7) The authors could say something more about the possible implications of their findings (if any) on their future research on this topic.

Reviewer #2: 1. Originality:

The authors have a good attempt to explore the attitude-behaviour gap in pro-environmental behaviours by comparing the contrasting effects of moral norms and idealism and relativism. The comparison of different predicting models is interesting and the strategies are novel and innovated. However, since the TPB model is relatively old model, the extensions of TPB model proposed by this paper the theoretical or methodological contributions are weak. Especially, the findings are difficult to be applied in real business context. The research models are well-established with strong theoretical support. The paper is generally well written and easy to follow. The mixed results have weakened the persuasiveness of the paper. Therefore, it is recommended to invite a revision, my decision will depend on how much you follow the comments & how much you improve the draft.

2. Relationship to Literature:

Generally, the paper demonstrated related theories and past literatures with adequate explanations and the research structure was developed on the foundation of closely-related previous findings. The research models were well-established with relatively strong theoretical support.

3. About Methodology and result: there are some concerns. The number of respondents and the number of participants included in the final analysis should be written in sequence, and the sections of participants, procedures, and data cleaning need to be integrated.

Usually, participants will receive allowance only after completing the entire questionnaire. The author of this article divided the questionnaire into six parts for payment- the way you did is relatively uncommon. I suggest you to cite the literature that was actually referenced.

All the constructs (measures) in this article should not be introduced separately, but integrated into the same paragraph. Please revise.

There are some issues in the result part. The current research needs to present coherent research result based on data analyses. It is recommended to include ALL variables in one model, using EFA for exploration and CFA for validating the exploratory results- just selecting some is uncommon- if you wanna do so, please justify with strong reasons.

4. Implications for research, practice and/or society:

The findings of the paper were interesting. For instance, the moral norms, rather than EPQ, was the predictor of attitudes; Idealism, rather than relativism, significantly predicted attitudes of all the seven behaviours when EPQ was placed as a predictor of attitudes. This study has extended the understanding of the effects of morality on consumers' pro-environmental intention and behaviour. However, due to the fact that this research is based on the TPB model, which is “kind-of” out-of-dated, the academic research contributions were weak. In addition the practical implications of this paper were not clearly described. In “Discussions” and “Conclusions”, there are no discussions about theoretical or methodological contributions. The authors need to give more compelling descriptions on the academic implications by addressing the findings about the attitude-behaviour gap.

5. Quality of Communication:

In general, the paper is well structured and well written. The major concepts are clearly delivered. The arguments were explained with the underlying logic. And the differences between proposed models were discussed with understandable explanations. However, some minor adjustments are needed in the paper. For instance, in "4. General Discussion", Page 40, the authors described about the paper's limitations. If a sub-title of "Limitations" is added before "Admittedly, the present study is limited in some key respects...", the structure would be clearer. In addition, some small modifications should be made to improve overall expressional quality. For example, in "4. General Discussion", Page 39, "the question of whether moral norms serve as a more accurate predictor of morality than idealism is more difficult." Please be reminded that the fluent expression will make it more reader-friendly. Further, Inconsistencies exist in the usage of quotation marks. For example, when attitude-behaviour gap was mentioned in the paper, the authors used "'", while the other quoted sentences used """. Adequate proofreading would make the work better.

6. PLOS authors have the option to publish the peer review history of their article (what does this mean?). If published, this will include your full peer review and any attached files.

Reviewer #1: No

Reviewer #2: No

---

## [Author Response · Author response to Decision Letter 0]

12 Jun 2023

Response to reviewer 1

1)To summarise, the sample of over a hundred of participants is enough for the SEM models to work (in fact, large sample sizes for SEM are not recommended), and the low participation rate does not mean that the responses of those participants who have completed all of the six studies were more invalid or more valid. Simply, the participants had no obligation to complete any of the further studies, and we have underestimated the efficiency of such an approach.

See response to Reviewer’ 2 comment, point 3.2

2)A standard way to test whether there is the difference between the constructs is to run EFA and CFA, which we have performed. The items that did not load on the correct factors where eliminated from the further analyses, making the EFA and CFA models statistically sound and establishing constructs as statistically distinct. Hence, the theoretical grounding and statistical analyses supports the idea that the constructs are distinct.

3)The following response added to the limitations section (p.38-39):

Furthermore, it’s unclear how much the scale measuring social norms is prone to measuring social desirability bias. Indeed, pro-environmental behaviours are highly desirable and social norms measure one’s adherence to desirable social norms. Tajfel’s theory of social identity and intergroup behaviour (Tajfel, 1974) suggests that people tend to maintain desirable social identity in the eyes of the others because evidence of behaving undesirably may contribute to the social identity threat (Branscombe et al., 1999; Scheepers & Ellemers, 2005). In turn, social identity threat may lead to perceiving one as an out-group rather than in-group member, and this status change, alongside individual’s rejection from the group can cause a fine level of inconvenience to individual’s everyday life. Hence, it’s difficult to tease out the measurement of social norms from measuring social desirability, rather than the actual level of individual’s adherence to social norms. For this reason, we have separated the survey into the six different studies believing that this approach might help reducing socially desirable responding. We believed this approach could be useful because the first survey measured one’s adherence to ten environmental behaviours (e.g. recycling, composting, etc.). Hence, we believed that not telling the participant about our intent to invite them to participate in the further surveys and leaving one month apart from the first and the second survey would be long enough for the participant to forget both our questionnaire items and their response to those. In addition, we have split the rest of the survey into five parts because the measurement scales are quite repetitive and we did not want to cause boredom to the participants. Hence, we preferred to provide five shorter questionnaires over the next upcoming days rather than one long questionnaire that the participant would be required to complete in one go. Such an approach has been previously suggested by Zaikauskaite and colleagues (2020) who have studied the relationship between morality and pro-environmental behaviours using the same ten pro-environmental behaviour items (Huang, 2016), and utilised by Zaikauskaite and colleagues (2022) who have also studied the relationship between morality and pro-environmental behaviours using the same ten pro-environmental behaviour items (Huang, 2016). In the latter study, the questionnaire was split into two parts where the survey measuring pro-environmental behaviour was launched one month prior to the rest of the survey, measuring the effectiveness of moral dimension using the GTME framework (Hunt & Vitell, 1986, 2006). Interestingly, the results of the study have demonstrated that GTME framework revealed the presence of the ‘attitude-behaviour’ gap. Hence, we attempted to replicate these findings using TPB model, same pro-environmental behaviour measurements (Huang, 2016), and methodological approach of separating pro-environmental behaviours from the rest of the survey items. Indeed, we have underestimated the efficiency of such an approach in terms of the difficulty to receive six completed parts of the survey coming the same participant, and yet, we do not have the ground to believe that such an approach somehow undermined the quality of the final responses because participants made no obligations to complete all the six parts of the survey (as intended, they did not know that six questionnaire parts exist).

4)Please see the response to your comment 3 above. 

5)We believe we have addressed this question in the responses to Reviewer 1 (point 1), Reviewer 1 (point3), and Reviewer 2 (point 3.2.). 

In general, smaller samples would not allow to run SEMs, as similar models require around 100 of responses, hence, having a smaller sample of ‘real life’ participants would not allow to run structural equations. Considerably, the study could be done using real life settings, but we did not have such a time allowance to do so with 100+ participants. However, it’s extremely interesting if a study conducted in real life setting would result in revealing no attitude-behaviour gap.

6)Edited:

Path analyses were performed to answer the research questions. First, we wanted to figure out whether the addition of a moral component to the TPB models composed of attitudes, perceived behavioural control, and intention-behaviour measurement items increases their predictive power (RQ1).

7)We have added section 5. Implications and future directions (p.40-42)

Response to Reviewer 2

1) We are uncertain of reviewer’s reasoning that the presentation should be different. The paper represents the correct sequence of what data we had from the beginning and the steps how it were analysed. We have followed the style of the similar recently published papers (among many others), e.g.: 

Chan, L., & Bishop, B. (2013). A moral basis for recycling: Extending the theory of planned behaviour. Journal of Environmental Psychology, 36, 96-102.

Zaikauskaite, L., Chen, X., & Tsivrikos, D. (2020). The effects of idealism and relativism on the moral judgement of social vs. environmental issues, and their relation to self-reported pro-environmental behaviours. Plos one, 15(10), e0239707.

Zaikauskaitė, L., Butler, G., Helmi, N. F., Robinson, C. L., Treglown, L., Tsivrikos, D., & Devlin, J. T. (2022). Hunt–Vitell’s General Theory of Marketing Ethics Predicts “Attitude-Behaviour” Gap in Pro-environmental Domain. Frontiers in psychology, 13.

2) To recap, neither MTurk or Prolific platforms offer an opportunity to recruit same participants for the separate studies and provide the compensation after the completion of all of the studies. Hence, after the participants have completed the first part of the questionnaire, we have used their ID to invite them to the further studies. Whether on not they wanted to participate further was a completely voluntary choice. Clearly, we did not expect that there will be such a low participation rate with regards to proceeding to the further studies, but this does not mean that the responses for the participants who have completed the questionnaire were invalid or more valid. Simply, low participation rate meant that the participant did not want to complete the next suggested study, and they had no obligations to do so. Of course, it would have been much more efficient if MTurk, Prolific or etc. platforms offered an option to recruit participants for the multiple studies and provide the payment after the completion of the last one. However, no such option exists. We have provided more explicit commentary on the reasoning to separate the questionnaire into six parts in the limitations section (see response to Reviewer 1, point 3).

We have added this explanation to section 5. Implications and Future directions

3)We are uncertain of the reviewer’s reasoning that our results are not coherent and are not based on the data analyses (what they are based on?), and reviewer’s recommendation to include ALL of the variables is also unclear, as the reviewer is not giving any references to base the comment on. And in fact, leaving ALL the variables in the model would not be statistically correct, as the acquired data would undermine EFA assumptions (items must load on different factors, otherwise the difference between the constructs can not be established). Certainly, it would have been better if all of the variables remained in the model, however, we do not see how excluding Social Norms would change the findings, as the aim was to assess whether or not ‘attitude-behaviour’ gap exists.

To recap: As we have stated in the manuscript, we aimed to perform the comparisons across all 5 of the models; however, after running EFA, we have discovered that some of the Social Norms items load incorrectly on the factors, meaning that it would be incorrect to proceed further with the CFA, SEM analyses for some of the models. According to the referenced statistical books and papers, it’s incorrect (not recommended) to proceed with the analyses when the factors do not load correctly. Hence, the only statistically correct way to make the models comparable is to take away the construct of the Social Norms from all of the models, and this is exactly what we have done to make the analysis coherent and statistically. In addition, in the supplementary materials we have provided EFA, CFA, and SEM of the original TPB models with and without Social Norms, demonstrating that the significance of the paths, model fit indices and etc. does not change when the Social Norms are excluded. Importantly, these models indicate that the prediction of the ‘attitude-behaviour’ gap remains the same regardless of whether or not Social Norms are excluded. As a reminder, our aim is to assess whether or not the model predicts ‘attitude-behaviour’ gap. Hence, taking or leaving Social Norms in the model does not really have an impact on meeting our research goals. Thus, we believe our analyses are coherent and correct, and we have previously provided this reasoning in the manuscript.

4) We do not believe it is accurate to say that the TPB model is “kind-of” outdated. Rather, TPB model is a well-established model that has perhaps been researched more than any other models in Social Psychology. Hence, our point was to show that ‘attitude-behaviour’ gap is tricky and complicated to assess because it might not always appear in the results, even at times when we are using well established models (“outdated) like TPB, measures (TPB measures and also different measures of morality), and procedures (e.g. statistical analyses like EFA, CFA, SEM). Hence, our study makes a contribution of raising the awareness to interpret positive results carefully, because these results might not be reflective of the real life situations (e.g. we know that ‘attitude-behaviour’ gap exists but the results say it does not exist). We have added section 5. Implications and Future directions

5)Sub-title ‘Limitations’ added to the manuscript. We have passed this new version of the manuscript to the professional proofreader.

---

## [Decision Letter · Decision Letter 1]

11 Jul 2023

PONE-D-22-33487R1The Theory of Planned Behaviour Doesn't Reveal 'Attitude-Behaviour' Gap? Contrasting the Effects of Moral Norms vs. Idealism and Relativism in Predicting Pro-Environmental BehavioursPLOS ONE

Dear Dr. Zaikauskaite,

Thank you for submitting your manuscript to PLOS ONE. After careful consideration, we feel that it has merit but does not fully meet PLOS ONE’s publication criteria as it currently stands. Therefore, we invite you to submit a revised version of the manuscript that addresses the points raised during the review process. Please submit your revised manuscript by Aug 25 2023 11:59PM. If you will need more time than this to complete your revisions, please reply to this message or contact the journal office at plosone@plos.org. Please include the following items when submitting your revised manuscript:A rebuttal letter that responds to each point raised by the academic editor and reviewer(s). You should upload this letter as a separate file labeled 'Response to Reviewers'.A marked-up copy of your manuscript that highlights changes made to the original version. You should upload this as a separate file labeled 'Revised Manuscript with Track Changes'.An unmarked version of your revised paper without tracked changes. You should upload this as a separate file labeled 'Manuscript'.If applicable, we recommend that you deposit your laboratory protocols in protocols.io to enhance the reproducibility of your results. Protocols.io assigns your protocol its own identifier (DOI) so that it can be cited independently in the future. For instructions see: https://journals.plos.org/plosone/s/submission-guidelines#loc-laboratory-protocols. Additionally, PLOS ONE offers an option for publishing peer-reviewed Lab Protocol articles, which describe protocols hosted on protocols.io. Read more information on sharing protocols at https://plos.org/protocols?utm_medium=editorial-email&utm_source=authorletters&utm_campaign=protocols.

We look forward to receiving your revised manuscript.

Kind regards,

Nicola Diviani

Academic Editor

PLOS ONE

Journal Requirements:

**Additional Editor Comments:**

Dear authors,

The original reviewers of the manuscript have now commented on the revised version and have found it to be substantially improved. As you can see below, Reviewer #2 still has a few minor points. When these will be addressed, the manuscript can be accepted for publication.

Best wishes

Reviewers' comments:

Reviewer's Responses to Questions

**Comments to the Author**

1. If the authors have adequately addressed your comments raised in a previous round of review and you feel that this manuscript is now acceptable for publication, you may indicate that here to bypass the “Comments to the Author” section, enter your conflict of interest statement in the “Confidential to Editor” section, and submit your "Accept" recommendation.

Reviewer #1: All comments have been addressed

Reviewer #2: All comments have been addressed

2. Is the manuscript technically sound, and do the data support the conclusions?

Reviewer #1: Yes

Reviewer #2: Partly

3. Has the statistical analysis been performed appropriately and rigorously? 

Reviewer #1: Yes

Reviewer #2: Yes

4. Have the authors made all data underlying the findings in their manuscript fully available?

Reviewer #1: Yes

Reviewer #2: Yes

5. Is the manuscript presented in an intelligible fashion and written in standard English?

Reviewer #1: (No Response)

Reviewer #2: Yes

6. Review Comments to the Author

Reviewer #1: The authors have addressed in detail all my previous comments, and in my opinion the manuscript is suitable for publication in PLoS ONE.

Reviewer #2: The authors have well addressed the issues mentioned in previous round, and problems in the previous review are mostly solved. I appreciate the authors took our comments and revised carefully. Besides, the authors have responded to the comments in the previous review in a detailed and nice manner- especially the authors significantly improved the arguments, literature review, and methodology. Moreover, the writing of the manuscript is updated substantially, and the quality of the manuscript is well improved.

In this sense, I recommend ‘Minor Revision’ to the paper after careful consideration and evaluation. If the authors can well improve the following 2 minor points I mentioned, I’ll recommend ‘Accept’ since the current draft is much improved than its previous version.

(1) Some updated and classic/related literature about TPB & Pro-environmental behavior were not mentioned/cited yet. Having symbolic and impactful literature review support will much enhance your arguments in no matter hypotheses development or theoretical/practical contribution. I suggest the author(s) to consider reading and citing the following updated literature; sure the authors can cite other similar papers not limit to listed ones.

-Mo. Z., Liu M., Liu, Y. (2018). Effects of functional green advertising on self and others, Psychology & Marketing. 35(5), 368-382.

-Wang, S., Liu, M.T. and Pérez, A. (2022), "A bibliometric analysis of green marketing in marketing and related fields: From 1991 to 2021", Asia Pacific Journal of Marketing and Logistics, Vol. ahead-of-print No. ahead-of-print. https://doi.org/10.1108/APJML-07-2022-0651

-Liu, M., Liu, Y., Mo, Z. (2020). Moral norm Is the key: An extension of the theory of planned behaviour (TPB) on Chinese consumers’ green purchase intentions, Asia Pacific Journal of Marketing and Logistics, 32(8), 1823-1841.

- Liu, Y., Liu, M., Pérez, A., Chan, W., Collado, J., Mo, Z. (2021). The importance of knowledge and trust for ethical fashion consumption, Asia Pacific Journal of Marketing and Logistics,33(5), 1175-1194.

(2) About Format, Tables & Figures demonstration:

Please revise the format based on publication standard/requirements set by Plos One before resubmission so it can save publisher's time to proceed your publicaiton.

7. PLOS authors have the option to publish the peer review history of their article (what does this mean?). If published, this will include your full peer review and any attached files.

Reviewer #1: No

Reviewer #2: No

---

## [Author Response · Author response to Decision Letter 1]

16 Aug 2023

Reviewer #1: No comments

Reviewer #2: 

Comment:

(1) Some updated and classic/related literature about TPB & Pro-environmental behavior were not mentioned/cited yet. Having symbolic and impactful literature review support will much enhance your arguments in no matter hypotheses development or theoretical/practical contribution. I suggest the author(s) to consider reading and citing the following updated literature; sure the authors can cite other similar papers not limit to listed ones. 

(1) -Mo. Z., Liu M., Liu, Y. (2018). Effects of functional green advertising on self and others, Psychology & Marketing. 35(5), 368-382. 

(2) -Wang, S., Liu, M.T. and Pérez, A. (2022), "A bibliometric analysis of green marketing in marketing and related fields: From 1991 to 2021", Asia Pacific Journal of Marketing and Logistics, Vol. ahead-of-print No. ahead-of- print. https://doi.org/10.1108/APJML-07-2022-0651

 (3)-Liu, M., Liu, Y., Mo, Z. (2020). Moral norm Is the key: An extension of the theory of planned behaviour (TPB) on Chinese consumers’ green purchase intentions, Asia Pacific Journal of Marketing and Logistics, 32(8), 1823-1841.

 (4)- Liu, Y., Liu, M., Pérez, A., Chan, W., Collado, J., Mo, Z. (2021). The importance of knowledge and trust for ethical fashion consumption, Asia Pacific Journal of Marketing and Logistics,33(5), 1175-1194.

Author's comment:

The focus of this article includes the review of TPB when (i) both intention and behaviour were measured using the TPB model, and (ii) either plain TPB, moral norms or EPQ were included. Hence, the suggested literature falls out of scope of this article:

(1)Does not include TPB

(2)Does not include TPB

(3)Does not include both intention and behaviour (model explores intention only)

(4)Does not include TPB

Reviewer's comment:

(2) About Format, Tables & Figures demonstration: Please revise the format based on publication standard/requirements set by Plos One before resubmission so it can save publisher's time to proceed your publicaiton.

Author's comment:

Thank you. We have edited the in-text statistical reporting, tables, and figures in the main body to adhere to Vancouver referencing.

---

## [Editor Report · Decision Letter 2]

17 Aug 2023

The Theory of Planned Behaviour Doesn't Reveal 'Attitude-Behaviour' Gap? Contrasting the Effects of Moral Norms vs. Idealism and Relativism in Predicting Pro-Environmental Behaviours

PONE-D-22-33487R2

Dear Dr. Zaikauskaite,

We’re pleased to inform you that your manuscript has been judged scientifically suitable for publication and will be formally accepted for publication once it meets all outstanding technical requirements.

Kind regards,

Nicola Diviani

Academic Editor

PLOS ONE
---

## [Editor Report · Acceptance letter]

22 Aug 2023

PONE-D-22-33487R2 

The Theory of Planned Behaviour Doesn't Reveal 'Attitude-Behaviour' Gap? Contrasting the Effects of Moral Norms vs. Idealism and Relativism in Predicting Pro-Environmental Behaviours 

Dear Dr. Zaikauskaitė:

I'm pleased to inform you that your manuscript has been deemed suitable for publication in PLOS ONE. Congratulations! Your manuscript is now with our production department. 

Kind regards, 

on behalf of

Dr. Nicola Diviani 

Academic Editor

PLOS ONE